# HIDRA 1.0: Deep-Learning-Based Ensemble Sea Level Forecasting in the Northern Adriatic

Lojze Žust[1], Anja Fettich[2], Matej Kristan[1], and Matjaž Ličer[3]

[1]Faculty of Computer and Information Science, Visual Cognitive Systems Lab, University of Ljubljana, Ljubljana, Slovenia
[2]Slovenian Environment Agency, Group for Meteorological, Hydrological and Oceanographic Modelling, Ljubljana, Slovenia
[3]National Institute of Biology, Marine Biology Station, Piran, Slovenia

**Correspondence:** Lojze Žust (lojze.zust@fri.uni-lj.si)

**Abstract.** Interactions between atmospheric forcing, topographic constraints to air and water flow, and resonant character of the basin make sea level modeling in Adriatic a challenging problem. In this study we present an ensemble deep-neural-network-based sea level forecasting method HIDRA, which outperforms our setup of the general ocean circulation model ensemble (NEMO v3.6) for all forecast lead times and at a minuscule fraction of the numerical cost (order of $2 \times 10^{-6}$).
HIDRA exhibits larger bias but lower RMSE than our setup of NEMO over most of the residual sea level bins. It introduces a trainable atmospheric spatial encoder and employs fusion of atmospheric and sea level features into a self-contained network which enables discriminative feature learning. HIDRA architecture building blocks are experimentally analyzed in detail and compared to alternative approaches. Results show the importance of sea level input for forecast lead times below 24 h and the importance of atmospheric input for longer lead times. The best performance is achieved by considering the input as the total sea level, split into disjoint sets of tidal and residual signals. This enables HIDRA to optimize the prediction fidelity with respect to atmospheric forcing while compensating for the errors in the tidal model. HIDRA is trained and analysed on a ten-year (2006-2016) timeseries of atmospheric surface fields from a single member of ECMWF atmospheric ensemble. In the testing phase, both HIDRA and NEMO ensemble systems are forced by the ECMWF atmospheric ensemble. Their performance is evaluated on a one-year (2019) hourly time series from tide gauge in Koper (Slovenia). Spectral and continuous wavelet analysis of the forecasts at the semi-diurnal frequency $(12\ \mathrm{h})^{-1}$ and at the ground-state basin seiche frequency $(21.5\ \mathrm{h})^{-1}$ is performed. The energy at the basin seiche in the HIDRA forecast is close to the observed, while our setup of NEMO underestimates it. Analyses of the January 2015 and November 2019 storm surges indicate that HIDRA has learned to mimic the timing and amplitude of basin seiches.

## 1 Introduction

Climate change is inducing sea level rise, adversely affects coastal ecosystems, economies and civil safety. In the shallow Northern Adriatic, low sea levels influence port activities and inhibit marine cargo while high sea levels cause substantial coastal flooding, inundation and erosion (Ferrarin et al., 2020), thus presenting a serious threat to Venice, Chioggia, Piran and other coastal towns and businesses in the region. Low sea levels predominantly occur when periods of high atmospheric pressure coincide with spring tide sea level minimums. High sea levels typically occur as storm surges during passages of

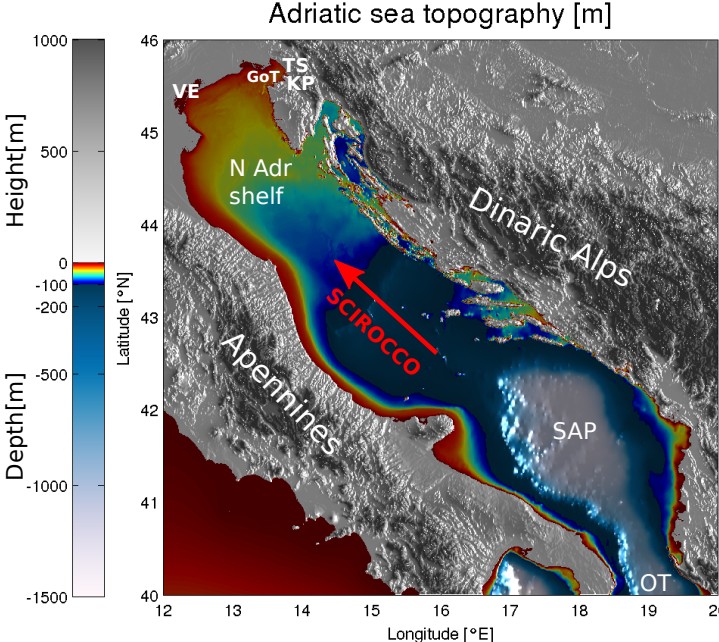

**Figure 1.** Adriatic orography and bathymetry. The following abbreviations are used: TS - Trieste, KP - Koper, GoT - Gulf of Trieste, VE - Venice, N Adr Shelf - Northern Adriatic Shelf, SAP - Southern Adriatic Pit, OT - Otranto Strait. Direction of Scirocco is marked with a red arrow. The image was created by the authors based on EMODnet bathymetry data, available at https://portal.emodnet-bathymetry.eu/ (last access: 27 October 2020) and Copernicus European Digital Elevation Model, available at https://land.copernicus.eu/imagery-in-situ/eu-dem/ eu-dem-v1-0-and-derived-products/eu-dem-v1.0 (last access: 27 October 2020).

atmospheric cyclones which manifest themselves as substantial air pressure lows and related winds over the basin. Reliable and timely sea level forecasting is thus a crucial element of early warning systems for mitigating the flooding consequences.

Adriatic Sea has an elongated basin with northwest-southeast orientation, lies in the Northern Central Mediterranean, and connects to the eastern Mediterranean basin through the Otranto strait at its southern end (see Figure 1). The basin lies embedded between the Alps (to the north), the Apennines (to the west) and Dinaric Alps (to the east) – it spans a 800 km long and 200

30   km wide area. The ridges significantly influence the basin circulation through topographic control of the air flow, especially during wind events of Bora (northeasterly wind) and Scirocco (southeasterly wind). Scirocco, predominantly directed along the basin long axis, is among the main drivers of the Adriatic storm surges. Northern Adriatic shelf is closed at its northern end and is the shallowest part of the Adriatic basin. Storm surges in this part are consequently most pronounced, causing substantial coastal flooding, inundation and erosion (Ferrarin et al., 2020).

The elongated basing results in seiches with a fundamental period of 21.5 h (and first excited mode period of 10.9 h) (Cerovecki et al., 1997; Medvedev et al., 2020). Diurnal and semi-diurnal tides enter the Adriatic through the Otranto Strait and force the basin close to its eigen-periods (Medvedev et al., 2020). Both seiches and tides are resonantly and topographically

amplified in the shallow northern Adriatic. In Northern Adriatic, crest-to-trough range of tidal sea level movement can reach up to 1.5 m (and up to 10 cm/s in current speed (Cosoli et al., 2013)). In absence of strong winds, this signal dominates the sea level dynamics. During storm surges however, tides are generally too weak to be dominant but remain a crucial part of the total sea level signal. Moreover, since Adriatic seiches decay on scales of several days, seiches and tides may continue to reinforce (or diminish) each other for several days.

The resonant character and high sensitivity to temporal phase lag between tides and seiches make Adriatic sea levels prediction a challenge for deterministic numerical ocean models. Even modest modeling errors in the timing, intensity or trajectory of an atmospheric cyclone, often lead to substantial modeling errors in the predicted sea level. Introduction of atmospheric numerical forecast ensembles has thus enabled implementation of operational sea surface height forecast systems that yield probabilistic forecasts along with error variance estimation, which show promise world-wide, see *i.e.* (Ferrarin et al., 2020; Bernier and Thompson, 2015; Mel and Lionello, 2014; Bertotti et al., 2011).

These systems, however, often involve a high computational cost, usually requiring running tens of runs of basin scale numerical ocean models (each forced by a different member of the atmospheric ensemble) each day (or even several times per day). Ensemble numerical modeling is therefore prohibitively demanding for many operational or civil rescue services that lack access to dedicated high-performance computing facilities.

Machine-learning-based ensemble modelling offers a possible solution to the challenges described above. Even though training a machine-learning model may involve substantial amount of training data and computational resources, the subsequent forecasting – even ensemble forecasting – is numerically cheap enough to be executed in some cases instantaneously on a standard personal computer. Early approaches employ classic machine learning methods (Imani et al., 2018) or shallow fully-connected neural networks (Pashova and Popova, 2011; Karimi et al., 2013) for daily or hourly sea level forecasting. The reported results show promise in sea level prediction, but fall short with simplistic architectures that ignore the atmospheric forcing. Ishida et al. (2020) attempt to improve the dynamics of hourly scale sea level forecasts by using a long short-term memory (LSTM) network, which are well established methods for sequence modelling and time-series prediction. The network considers several atmospheric variables (wind speed and direction, sea level pressure, air temperature) as well as relative positions of the Sun and the Moon and annual global air temperatures as the input. Empirically, the short-term sea level prediction of one hour in the future is improved compared to older approaches. While predictions farther into the future could be achieved by iterative auto-regression, the errors would likely exponentially increase. A longer prediction horizon is considered by Braakmann-Folgmann et al. (2017), who apply a combination of recurrent neural networks (LSTMs) and convolutional neural networks to model monthly spatial sea level time series. This is one of the first works that considers both spatial and temporal aspects of the input data, however, predictions are made at a much coarser temporal resolution than that considered in this paper. A higher temporal resolution is considered by Hieronymus et al. (2019), which is the most closely related work to ours. Autoregressive neural networks are used to model the sea level time-series with addition of atmospheric forcing reduced by empirical orthogonal function (EOF) decomposition. For northern Adriatic, Venice lagoon specifically, artificial neural networks have already been shown to be useful at modelling numerical model errors in sea level prediction (Bajo and Umgiesser, 2010).

In this work we propose HIDRA – a HIgh-performance Deep tidal Residual estimation method using Atmospheric data. HIDRA is a novel deep learning architecture which employs tidal and atmospheric forcing contributions for accurate sea level predictions. The model is trained end-to-end with discriminative feature extraction as part of the learning to maximize the forecast accuracy and to compensate for the inaccuracies of the astronomical tide estimates. HIDRA is benchmarked against the operational setup of NEMO v3.6 general circulation model engine (Madec, 2008), which is run daily as part of the National Hydrological Forecasting Service at the Slovenian Environment Agency (ARSO). For brevity, we refer to this particular setup (see the *Code and data availability* Section for a detailed configuration namelist) as *the NEMO model*.

The remainder of the paper is structured as follows. The input sea level and atmospheric data, and the datasets used in this study are described in Section 2. The HIDRA architecture, our NEMO ocean model setup and the ensemble structure are presented in Section 3. Section 4 reports a detailed analysis of the HIDRA architecture and empirical comparison to the NEMO system on a challenging setup. Conclusions are drawn in Section 5.

## 2 Sea Level and Atmospheric Data

### 2.1 Sea Level Data

Sea surface height (SSH) measurements were obtained from the Koper Mareographic Station ($45°33'$ N, $13°44'$ E; see Figure 2 for location), which is operated by the Slovenian Environment Agency (ARSO). The measurements are acquired by a bottom-mounted pressure gauge in ten minute intervals, which are subsequently quality controlled at ARSO. The tidal part of the sea level is independent from atmospheric forcing and can be approximated by tidal models. We analyzed the tidal contribution to Koper SSH using the Tidal Analysis Program for Python TAPPY (Cera, 2011). Tidal contribution is estimated from a 20-year hourly time-series of Koper SSH for the period between January 1995 and September 2014. Tidal constituents are then used to estimate past and future tidal values. The residual sea level is defined in this paper as the arithmetic difference between the total and the tidal sea level.

### 2.2 Atmospheric Data

Atmospheric data is obtained from Ensemble Prediction System (EPS) of the European Centre for Medium-Range Weather Forecasts (ECMWF). The data comes as an ensemble of fifty integrations of global atmospheric models (Leutbecher and Palmer, 2007). Ensemble forecasts have a 0.125° arc degree spatial (zonal and meridional) resolution and a 3-hour temporal resolution. In this study, the following forecast fields were subset to the Adriatic basin, represented by a $73 \times 57$ spatial grid (see Figure 2): (i) 10-meter zonal and meridional winds, (ii) mean sea level pressure and (iii) air temperature at 2 metres. The forecasts were linearly interpolated to hourly timesteps to match the SSH temporal resolution. Atmospheric fields over land and sea are treated in the same manner, *i.e.* while HIDRA does receive an explicit spatial encoding of atmospheric fields (Section 3.1.1), it does not employ a land/sea mask.

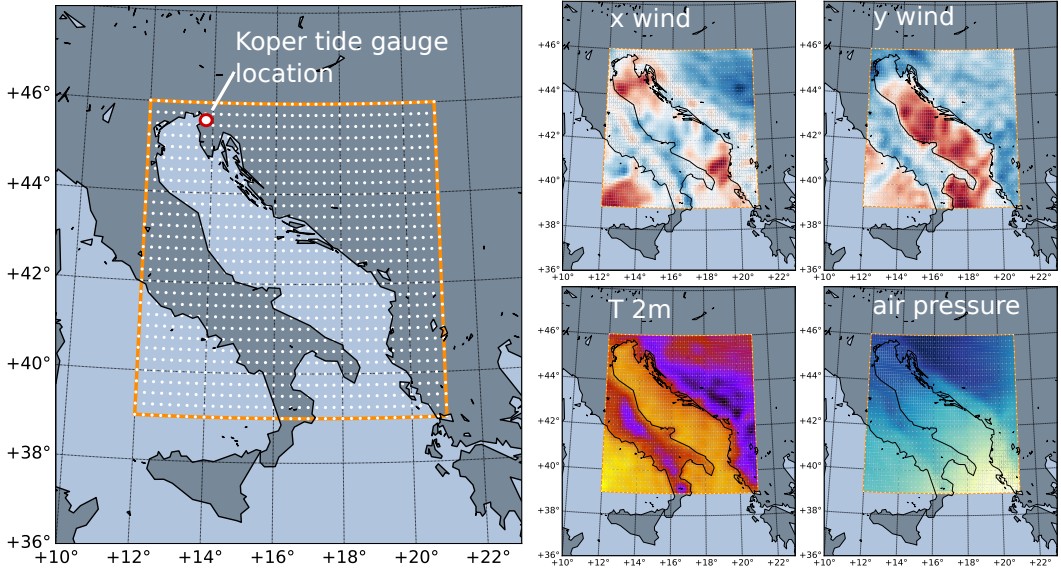

**Figure 2.** Left: NEMO ocean model domain (orange rectangle) and ECMWF ensemble grid points (white dots). Every second ECMWF grid point is displayed for clarity. Koper tide gauge location is marked with red-white circle. Right: four ECMWF atmospheric fields are extracted over the region at each time-step: zonal (x) and meridional (y) wind, air temperature at 2 meters and surface air pressure.

Direct wind influence on the ocean is exerted through vertical momentum transfer or wind stress. Configurations with both raw wind and wind stress were tested to obtain optimal neural network configuration. Whenever wind stress was used, turbulent
momentum transfer (wind drag) coefficient was computed using the Large and Pond parametrization (Large and Pond, 1981). Note that the purpose of wind stress parametrization in this study is not to most concisely represent the vertical momentum flux at ocean surface (which would require more complex schemes), but merely to introduce to the neural network the nonlinear wind stress dependence on the wind to assist its learning process.

### 2.3  Evaluation datasets

Atmospheric and sea level data described in previous sections were used to create a dataset for years 2006–2016. The first 80% of the dataset is used for training (70%) and validation (10%), while the last 20% (September 2014 - December 2016) is used for testing. The data is standardised and global average pooling is used to reduce the dimensionality of the atmospheric data – spatial dimension of the data in samples is reduced in half, from $73 \times 57$ points to $37 \times 29$ points, and the temporal dimension of atmospheric data is reduced by a factor of 4. Sea level data retains 1-hour resolution. An additional test-only dataset for the
115  year 2019 was constructed in the same manner and was used for comparison of our setup of NEMO and HIDRA prediction performance.

Oversampling is applied to the training data to improve the prediction accuracy of the rare storm surge events. The training dataset is split into two subsets by thresholding the residuals at $40\,\mathrm{cm}$. Storm surges (residual > $40\,\mathrm{cm}$) represent approximately two percent of the data. During training the samples are randomly sampled from each of the two subsets with equal probability.

## 3  Numerical Models

### 3.1  HIDRA

HIDRA (a HIgh-performance Deep tidal Residual estimation method using Atmospheric data) is a deep neural network that predicts future surface height residual values (relative to the tidal model) from the sea level and atmospheric forcing input tensors. The atmospheric state at time-step $t$ is represented by an atmospheric tensor $\mathbf{I}_t \in \mathcal{R}^{H \times W \times 4}$, where $W = 37$ and $H = 29$ are the numbers of domain cells in longitudinal and latitudinal direction, respectively. The third dimension of $\mathbf{I}_t$ corresponds to the four atmospheric surface input fields: two components of the wind stress, mean sea level pressure and air temperature at $2\,\mathrm{m}$ (see Figure 2).

To account for the causal relation between past atmospheric and tidal forcing in the basin and future sea surface heights, HIDRA considers the forcing data over a range of past and future timesteps. In particular, for prediction starting at time $t_0$, HIDRA takes as the input atmospheric tensors $\mathbf{I}_t$ for the interval $t \in [t_0 - T_{\min} + 1, t_0 + T_{\max}]$ and the tidal and the residual values from the interval $t \in [t_0 - T_{\min}, t_0]$, and predicts residual values for the interval $t \in [t_0 + 1, t_0 + T_{\max}]$. Here, $T_{\min}$ defines the number of past hours considered in sea level prediction and $T_{\max}$ denotes the prediction horizon. In our experiments the predictions are made for 72 hours into the future, thus $T_{\max} = 72\,\mathrm{h}$ and we have determined (see Section 4.1.1) that extending the historical horizon beyond 24 hours does not affect the prediction accuracy, thus we set $T_{\min} = 24\,\mathrm{h}$. Note that the atmospheric tensor contains future forecasts as well, while the input sea level vectors contain only tides and residuals observed up to the prediction time $t_0$.

The HIDRA architecture is summarized in Figure 3. Atmospheric tensors from all considered time-steps are individually encoded by an *atmospheric spatial encoder* (ASE) module (Section 3.1.1) and fused by the temporal encoder block (Section 3.1.2) based on the temporal attention mechanism into an atmospheric feature vector. The resulting vector is concatenated with the past tidal and residual measurements. This is followed by a *residual regression block* (Section 3.1.3) to generate the final residual predictions $\hat{\mathbf{r}}_t$ along with their uncertainties $\boldsymbol{\sigma}_t$.

### 3.1.1  Atmospheric Spatial Encoder

The atmospheric spatial encoder (ASE) encodes the spatially-represented atmospheric data into features, fine-tuned for the task of sea level prediction, i.e., the atmospheric tensor $\mathbf{I}_t \in \mathcal{R}^{29 \times 37 \times 4}$ for time-step $t$ is encoded into a feature vector $\mathbf{f}_t \in \mathcal{R}^{256 \times 1}$. The ASE architecture (shown in Figure 3a) follows design principles of the ResNet20 v2 convolutional neural network (He et al., 2016), which has already demonstrated remarkable performance in image processing tasks. ASE is composed of 22 convolutional layers. Spatial dependence in feature extraction in enforced by concatenating the atmospheric tensor $\mathbf{I}_t$ with a

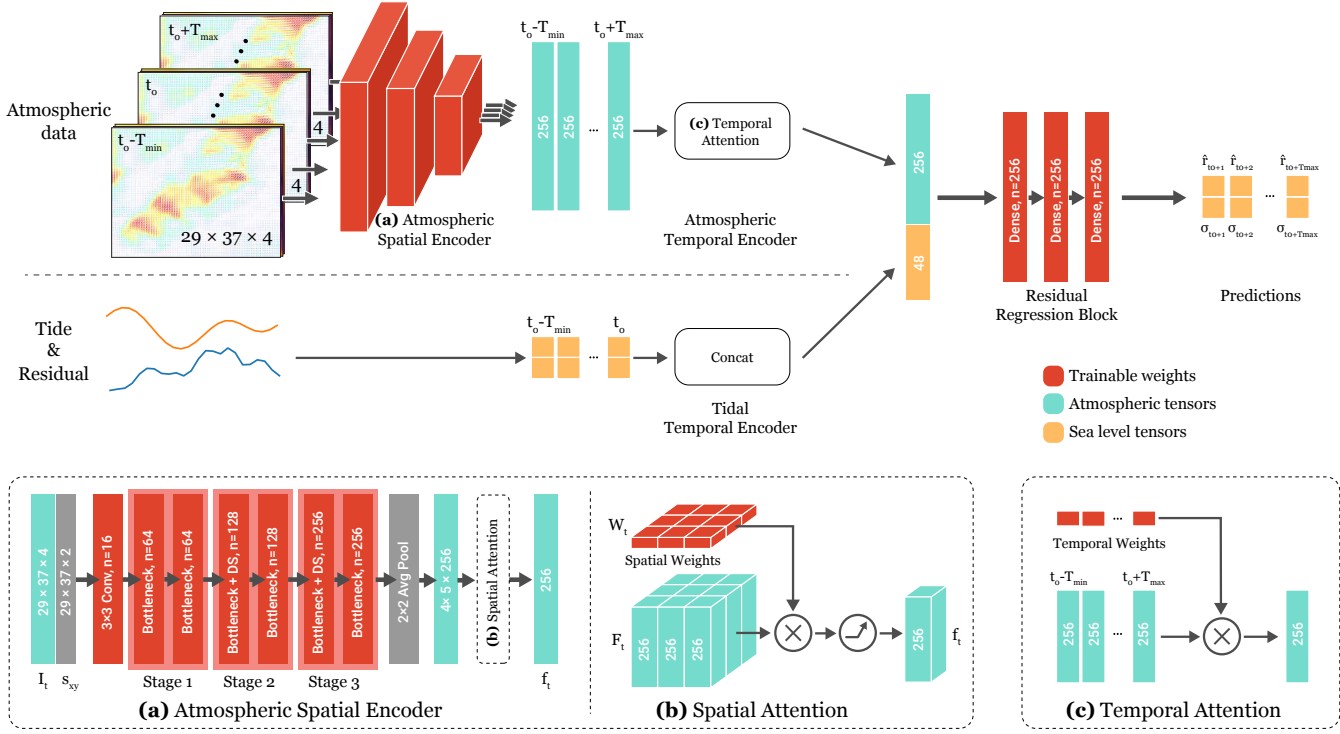

**Figure 3.** The proposed HIDRA architecture. A convolutional *Atmospheric Spatial Encoder* (ASE) extracts spatial atmospheric features from each time-step. Atmospheric and sea level temporal features are encoded by respective Temporal Encoder blocks, fused and passed to the fully-connected *Residual Regression Block* to predict the residuals along with their uncertainties. With $n$ we denote the number of filters or units of the block. The trainable blocks are colored red. The structure of the bottleneck blocks used in the ASE is presented in Figure 4.

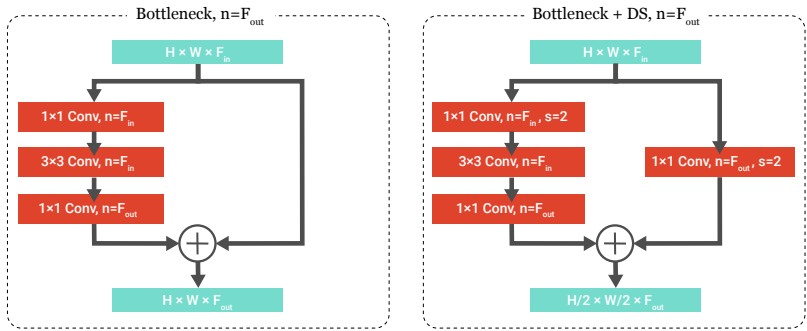

**Figure 4.** Structure of the bottleneck blocks used in the Atmospheric Spatial Encoder (Figure 3). The bottleneck block takes a feature map with depth $F_{\text{in}}$ as the input and outputs a feature map with depth $F_{\text{out}}$. A regular bottleneck block (left) retains the spatial dimensions of the feature maps, while the downsampling (DS) bottleneck block (right) uses strided convolutions to reduce the spatial dimensions in half. We denote the number of convolutional filters by $n$ and the stride parameter by $s$.

spatial encoding tensor $\mathbf{s}_{xy} \in \mathcal{R}^{29 \times 37 \times 2}$, which contains the $x$ and $y$ coordinates (scaled between 0 and 1) of each pixel in the tensor. The augmented tensor is then processed by a convolutional layer with 16 filters of $3 \times 3$ size, followed by three ResNet20 v2 stages, a spatial pooling layer and a time-dependent spatial attention layer (Figure 3b).

Each ResNet stage consists of two residual (bottleneck) blocks, where each block contains three convolutional layers (*i.e.*, $1 \times 1$, $3 \times 3$ and $1 \times 1$) and a residual connection, which sums the block input with its output. To match the number of output features in the residual connection, the first residual block in each stage uses an additional $1 \times 1$ convolutional projection. Spatial feature downsampling (DS) is applied by a stride of length 2 in the first convolutional layer of the second and third stage to increase the effective receptive field of neurons. A ReLU activation layer is appended to each convolutional layer, except from the first one, are pre-pended by a batch-normalization layer to stabilize the learning.

The output of the last residual block is spatially reduced by half with an average pooling layer, resulting in a feature tensor $\mathbf{F}_t$ of size $5 \times 4 \times 256$. Finally, a time-dependent spatial attention layer produces the final feature vector $\mathbf{f}_t \in \mathcal{R}^{1 \times 256}$, which is a weighted sum of spatial positions

$$\mathbf{f}_t = \mathrm{ReLU}\left( \sum_{i=1}^{4} \sum_{j=1}^{5} \mathbf{F}_t^{(i,j)} \mathbf{w}_t^{(i,j)} \right), \tag{1}$$

where $\mathbf{F}_t^{(i,j)}$ is a slice of the feature tensor at time $t$ at spatial coordinates $(i,j)$, $\mathbf{w}_t^{(i,j)}$ is the respective spatial weight and $\mathrm{ReLU}(\cdot)$ is the ReLU activation function. Note that the weight matrices are temporally dependent, which allows them to focus on different parts of the atmospheric feature maps over time. With the exception of the spatial attention layer, all weights of the ASE network are temporally-independent and are thus shared between all atmospheric input tensors.

### 3.1.2 Temporal Encoders

The ASE encodes input sequence of atmospheric tensors into a sequence of atmospheric features. These are stacked into an atmospheric feature matrix $\mathbf{F} \in \mathcal{R}^{256 \times T_\Delta}$, where $T_\Delta$ is the number of time-steps in the atmospheric input tensor $\mathbf{I}_t$, and compressed into a single feature vector $\mathbf{f} \in \mathcal{R}^{256 \times 1}$ by a weighted summation

$$\mathbf{f} = \mathbf{F}\mathbf{w}^T, \tag{2}$$

where $\mathbf{w}$ is a temporal weights vector which serves as a temporal attention mechanism (Figure 3c) and adjusts the contributions of different past time-steps to maximize the prediction performance.

The input tidal and residual sequences, each a $T_{\min} \times 1$ vector, are concatenated with the encoded atmospheric feature vector (2) into the combined temporally-encoded atmospheric and surface height feature vector. This vector is passed to the residual regression block (Section 3.1.3) for the final prediction.

### 3.1.3 Residual Regression Block

Probabilistic regression is employed to enable predicting the most likely residual values along with their uncertainties. For each time-step, the mean and variance of a Gaussian probability density function are predicted. The residual regression block

output are thus two sequences $\hat{\mathbf{r}}$ and $\hat{\boldsymbol{\sigma}}$, each a $1 \times T_{\max}$ vector, where $T_{\max}$ is the prediction horizon. The residual regression block (Figure 3) is composed of three dense fully-connected layers, each consisting of 256 units, followed by a fully-connected layer that maps into two $1 \times T_{\max}$ vectors for the means $\hat{\mathbf{r}}$ and standard deviations $\hat{\boldsymbol{\sigma}}$. A soft-plus function (Glorot et al., 2011) is applied to the standard deviation vector to ensure positive values.

### 3.1.4 Network Training

A loss function that takes into account the probabilistic outputs is designed for training HIDRA end-to-end. In particular, a log-likelihood of the ground-truth data (*i.e.*, the residuals series $\mathbf{r}_{t_0} = \{r_{t_0+\delta}\}_{\delta=0:T_{\max}}$), is computed under the sequence of predicted Gaussians (Figure 3, $\hat{\mathbf{r}}_{t_0}$ and $\hat{\boldsymbol{\sigma}}_{t_0}$) for all training sequences starting at different times $t_0 \in \mathcal{T}$. The loss is thus defined as

$$\mathcal{L}(\mathbf{r}, \hat{\mathbf{r}}, \hat{\boldsymbol{\sigma}}) = -\sum_{t_0 \in \mathcal{T}} \sum_{\delta=0}^{T_{\max}} \log \left( \frac{1}{\hat{\sigma}_{t_0+\delta}\sqrt{2\pi}} \exp\left( (r_{t_0+\delta} - \hat{r}_{t_0+\delta})^2 / \hat{\sigma}^2_{t_0+\delta} \right) \right), \tag{3}$$

where $\mathbf{r}$ are the sets of training samples, while $\hat{\mathbf{r}}$ and $\hat{\boldsymbol{\sigma}}$ are the corresponding HIDRA predictions and their error estimates, respectively.

The network and experiments are implemented using Google's machine learning library TensorFlow (Abadi et al., 2015). We use the ADAM optimizer with learning rate of 0.001, $\beta_1 = 0.9$ and $\beta_2 = 0.999$ to train the model. The training batch size is set to 64 data samples. The models are trained for 30 epochs, each epoch consisting of 1000 training steps (batches). A computer with a NVIDIA GeForce GTX 980 graphics card was used for model training and evaluation.

## 3.2 NEMO Ocean Model

General circulation model NEMO v3.6 (Madec, 2008) is used as a baseline for comparison with HIDRA. Detailed configuration namelist of our particular setup described below is available in the supplementary material in the paper.

Adriatic NEMO model used in this study is set up on a regular longitude-latitude grid (648×504 cells) with a 1°/72 arc-degree horizontal resolution and 31 vertical partial step $z^*$-levels. The model domain spans 12–21° E and 39–46° N (see Figure 2). In all regions shallower than 2 m, a 2 m depth is enforced. Baroclinic timestep was set to 120 s. Barotropic timestep is adjusted to meet Courant-Friedrichs-Lewy stability condition. This operational suite runs every day at Slovenian Environment Agency (ARSO) High Performance Computing Center and is hotstarted from the run of the previous day. Hourly lateral boundary conditions in the Ionian Sea are taken from the hourly Copernicus CMEMS Mediterranean Sea Analysis and Forecast product. Turbulent momentum and heat fluxes across the air-sea interface are parametrized using CORE bulk flux formulation (Large and Yeager, 2004) using ECMWF ensemble atmospheric fields.

Rivers are modeled as discharge of fresh water at the respective river location as described in Ličer et al. (2016). Flather boundary condition determines barotropic dynamics at the lateral open boundary, while Flow Relaxation Scheme (Engedahl, 1995) is applied for baroclinic dynamics and tracers. Lateral momentum boundary condition at the coast is free-slip. Bottom boundary layer is logarithmic with nonlinear bottom friction. Lateral diffusion is governed by Laplacian operators for tracers

and dynamics, both operating over geopotential surfaces. Generic Length Scale k-$\epsilon$ scheme is used for vertical diffusion. Sur-
face wave mixing is parametrized using Craig and Banner formulation (Craig and Banner, 1994). The full NEMO configuration namelist is provided as supplementary material (Žust et al., 2021).

NEMO was run in this study without tidal forcing and predicts the residual sea level for the entire Adriatic basin with the forecast period set to 72 h (as in HIDRA). In the ensemble simulations, only eight out of fifty ECMWF ensemble members were used as forcing to our NEMO circulation model due to computational constraints. These eight ensemble members were

selected from ECMWF ensemble based on the wind strength each member exhibits in the central Adriatic: ECMWF ensemble members were ordered by wind strength in the central Adriatic and then a subset of members was made from the strongest to the weakest member in steps of 6 (*i.e.* integer part of $50/8$). This generates eight possible forcing scenarios in the Adriatic basin while conserving the wind forecast spread of the reduced ensemble. For the $i$-th ($i = 1, \ldots, 8$) NEMO ensemble member run, residual sea level forecast for Koper is extracted from the NEMO basin prediction as a single time-series. This is then

added to the tidal time series, obtained via tidal analysis from observations, to obtain the total modeled sea level, which we denote as $y^{\mathrm{nemo}}(i,t)$.

Each member of the NEMO ensemble sea level forecast for Koper is further corrected for bias. This is necessary to compensate for the fact that NEMO sea level reflects departures from a local geoid and does not represent the absolute local depth of the water, which is also driven by low-frequency processes (like planetary waves in air pressure), which cannot be reflected in

the 72 hour run in a regional basin. To obtain the absolute sea level needed by port and civil rescue authorities, the NEMO sea level predictions have to be adjusted to the Koper tide gauge observations.

On the $n$-th hour of the forecast day, the model bias with respect to Koper tide gauge observations $y^{\mathrm{kp}}(t)$ can is estimated as

$$\epsilon_n(i) = n^{-1} \sum_{k=1}^{n} \left[ y^{\mathrm{nemo}}(i,t_k) - y^{\mathrm{kp}}(t_k) \right]. \tag{4}$$

The $i$-th ensemble NEMO prediction time-series $y^{\mathrm{nemo}}(i,t)$ is then shifted by $\epsilon_n(i)$ so that the bias of the first $n$-hours of the $y^{\mathrm{nemo}}(i, 1 < t < n)$ with respect to observations is zero. Complete forecast time-series $y^{\mathrm{nemo}}(i,t)$ will of course still exhibit a non-zero bias. This procedure is applied every hour as new observations from Koper tide gauge arrive. Note that, unlike NEMO ensemble, HIDRA ensemble does not need any such bias correction, because it already contains local tide gauge information through its sea level input in the day prior to the forecast and learns to adjust for the possible bias.

For operational reasons, first daily NEMO sea level ensemble run usually becomes available between 11 00 and 12 00 UTC at the earliest. This means that the earliest bias correction of each day generally takes into account the first 12 hours of tide gauge observations of that day. Raw NEMO time-series from $i$-th ensemble member $y^{\mathrm{nemo}}(i,t)$ therefore gets shifted by $\epsilon_{12\mathrm{h}}(i)$ to produce the first bias-corrected forecast of that day, i.e.,

$$y_{\mathrm{bc12h}}^{\mathrm{nemo}}(i,t) = y^{\mathrm{nemo}}(i,t) - \epsilon_{12\mathrm{h}}(i), \tag{5}$$

with $\epsilon_{12\mathrm{h}}(i)$ defined in (4). The bias-corrected NEMO ensemble mean, ensemble maximum and ensemble minimum time-series is then constructed from 12-hour bias-corrected ensemble members at each forecast timestep in an identical fashion as with HIDRA – see (6)-(8). The corresponding time-series are denoted as $\overline{y}_{\mathrm{bc12h}}(t)$, $y_{\mathrm{bc12h}}^{\max}(t)$ and $y_{\mathrm{bc12h}}^{\min}(t)$.

### 3.3 Ensemble Statistics

As mentioned in Section 2.2, a total of fifty ECMWF atmospheric ensemble members are available daily. This results in an ensemble of $n_{ens} = 50$ sea level forecasts by HIDRA and an ensemble of $n_{ens} = 8$ sea level forecasts by NEMO. The ensemble mean time-series is defined as the average over all ensemble members predictions

$$\overline{y}(t) = n_{ens}^{-1} \sum_{i=1}^{n_{ens}} y(i,t), \tag{6}$$

where $y(i,t)$ is the $i$-th member. Similarly, the ensemble prediction envelope, i.e., the per-time-step minimum and maximum sequence, is defined as

$$y^{\max}(t) = \max_i[y(i,t)], \tag{7}$$
$$y^{\min}(t) = \min_i[y(i,t)]. \tag{8}$$

In the interest of clarity, only ensemble means, maximums and minimums, as defined above, are analyzed in the following (rather than individual ensemble members).

## 4 Results and Discussion

Predictions from HIDRA and NEMO are discussed in two sections. Section 4.1 analyzes the influence of atmospheric and sea level input on HIDRA forecasts and concludes with a brief analysis of HIDRA atmospheric encoder design. Statistical and spectral analyses of HIDRA and NEMO predictions are then presented in Section 4.2.

### 4.1 HIDRA Architecture Analysis

The HIDRA architecture design choices and their impact on forecast accuracy is analyzed in this section. Forecast accuracy is tested (i) with regard to the prediction lead time, i.e., the number of hours in the future we are forecasting, and (ii) with regard to the sea level residual value, i.e., how far away from astronomical tide lies the sea level. All experiments regarding network design are performed on the Koper test sea level dataset, which spans November 2014 to December 2016.

Influence of the historic horizon is examined in Section 4.1.1. Contribution of individual data sources (*i.e.* atmospheric data and sea level history data) is analyzed in Section 4.1.2. The influence of the residual forecasting approach is evaluated in Section 4.1.3. The influence of the proposed atmospheric data encoder in comparison to reconstruction-based empirical orthogonal functions (EOF) is examined in Section 4.1.4, while the influence of the temporal encoder is analyzed in Section 4.1.5.

**Table 1.** HIDRA performance for different historic horizons in terms of mean absolute error (MAE), root mean squared error (RMSE), model bias and likelihood shown separately on all data and on storm surge events. CPU execution time (on a single core) per example is also reported.

| | MAE [cm] | RMSE [cm] | Bias [cm] | Likelihood | CPU time [s] |
|---|---|---|---|---|---|
| **Overall** | | | | | |
| $T_{min} = 12$ | 5.3 | 7.0 | **-0.2** | 0.0440 | **0.17** |
| $T_{min} = 24$ | 4.9 | **6.4** | -0.4 | **0.0470** | 0.19 |
| $T_{min} = 36$ | **4.8** | **6.4** | -0.9 | 0.0455 | 0.21 |
| $T_{min} = 48$ | **4.8** | **6.4** | -0.6 | 0.0438 | 0.23 |
| **Storm surge events** | | | | | |
| $T_{min} = 12$ | 11.7 | 13.9 | -11.2 | 0.0220 | **0.17** |
| $T_{min} = 24$ | **10.3** | 12.9 | -9.3 | **0.0253** | 0.19 |
| $T_{min} = 36$ | 10.7 | 13.2 | -9.8 | 0.0245 | 0.21 |
| $T_{min} = 48$ | 10.4 | **12.8** | **-9.1** | 0.0251 | 0.23 |

### 4.1.1 Influence of the Historic Horizon

We first analyze the influence of the HIDRA historic horizon defined by the parameter $T_{min}$ (see Section 3.1). Table 1 summarizes the performance of HIDRA with $T_{min} \in \{12, 24, 36, 48\}$, which translates to historic horizons of 12, 24, 36 and 48 hours prior to the beginning of forecast. Increasing the historic horizon from 12 to 24 hours significantly improves the prediction accuracy (9% reduction in RMSE error), however, further increases of the historic horizon (i.e., to 36 or 48 hours) do not show measurable benefits. Note that the execution time increases with the length of the historic horizon due to a substantial increase of parameters on the input layer. For this reason, we use a historic horizon of 24 hours ($T_{min} = 24$) as the best trade-off in the remaining analysis and denote this version as HIDRA$_0$.

### 4.1.2 Contribution of Atmospheric and Sea Level Inputs

HIDRA uses two input sources: the atmospheric data and the sea level history. In this study we analyze the individual contributions of both input sources. The full HIDRA model (using both atmosphere and sea level data input, denoted as HIDRA$_0$) is compared with two single-input-source models: (i) HIDRA$_{Ai}$, using only atmospheric inputs, and (ii) HIDRA$_{SLi}$, using only sea level inputs. In both setups, the network branch responsible for processing the ignored input source is removed. Results are presented in Table 2 and Figure 5.

Table 2 indicates that HIDRA exhibits best performance when using both atmospheric and sea level input and outperforms both single-source variants by a large margin. This holds overall and also within limited time windows during storm surge events (defined as the timestamps where the residual is larger than 40 cm). Removing each source individually leads to a significant performance drop. The RMSE increases by 77% when using only the atmospheric input data (HIDRA$_{Ai}$), while a 83% RMSE increase is observed when using only sea level data (HIDRA$_{SLi}$). This confirms that both input data sources are

**Table 2.** HIDRA performance for individual input data sources in terms of mean absolute error (MAE), root mean squared error (RMSE) and model bias. Full HIDRA$_0$ with both input sources is compared with alternatives that use only atmospheric (HIDRA$_{Ai}$) or tidal forcing (HIDRA$_{SLi}$) as the input. Performance for the atmospheric tide is provided as reference. Performance is reported on all data as well as on storm surge events only.

|  | MAE [cm] | RMSE [cm] | Bias [cm] | Likelihood |
|---|---|---|---|---|
| **Overall** |  |  |  |  |
| HIDRA$_0$ | **4.9** | **6.4** | **-0.4** | **0.0470** |
| HIDRA$_{Ai}$ | 8.8 | 11.3 | -0.5 | 0.0315 |
| HIDRA$_{SLi}$ | 8.6 | 11.7 | 3.3 | 0.0279 |
| Reference (tide) | 12.1 | 15.7 | -2.4 | – |
| **Storm surge events** |  |  |  |  |
| HIDRA$_0$ | **10.3** | **12.9** | **-9.3** | **0.0253** |
| HIDRA$_{Ai}$ | 20.1 | 22.5 | -19.5 | 0.0093 |
| HIDRA$_{SLi}$ | 21.1 | 25.5 | -20.2 | 0.0134 |
| Reference (tide) | 49.6 | 50.4 | -49.6 | – |

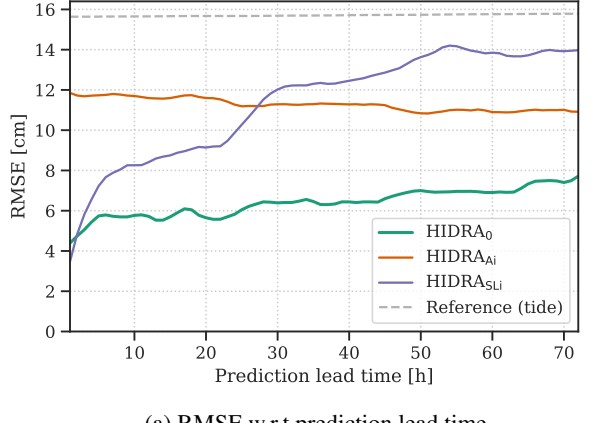

(a) RMSE w.r.t prediction lead time

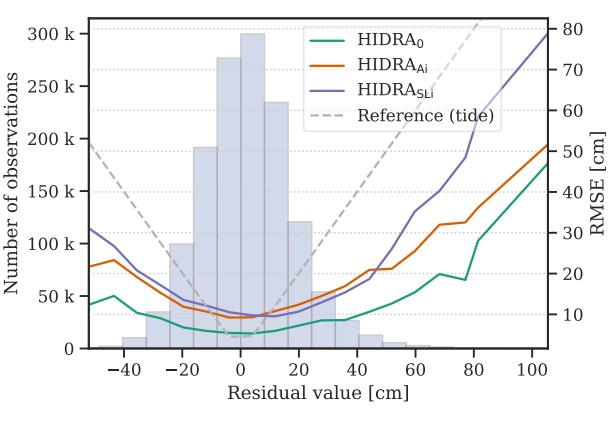

(b) RMSE w.r.t. residual value

**Figure 5.** Performance of single-input-source models. Root Mean Squared Error (RMSE) with respect to (a) prediction lead time and (b) residual value bins for different input sources is visualized. The full HIDRA model is compared with single-input-source variants – atmospheric data only HIDRA$_{Ai}$ model and tidal data only HIDRA$_{SLi}$ model. The RMSE of the astronomic tide is shown for reference.

essential for accurate prediction. Note also that both input data sources have a similar overall contribution to the prediction accuracy.

Performance analysis over prediction lead time (Figure 5a) reveals further insights. $HIDRA_{Ai}$ is very consistent across the entire prediction interval. In fact, the error slightly decreases over time (8% decrease in RMSE over the interval of 72 h). $HIDRA_{SLi}$, on the other hand, is a much better sea level predictor in the short-term, but the error increases rapidly over the prediction interval (by 400% over the interval of 72h). Thus, while the sea level data is important for short-term predictions, the atmospheric data is more informative for predictions further into the future.

Grey dashed line in Figure 5 depicts forecast errors of using the astronomical tide values as the surface height predictor. Since tidal forecast is done independently of prediction lead time, its RMSE over prediction time window is simply a root mean square error of the tidal model, plotted as a horizontal line. Bias of the reference tidal model with regard to specific residual bin is, by definition, the negated value of the residual itself, while its RMSE is, again by definition, simply the absolute value of the residual itself. Whenever HIDRA exhibits lower biases or RMSEs than tidal reference model, they are essentially predicting more accurately than a tidal model would. Note that this is the case for all prediction lead times and practically all residual values.

On large residual values that correspond to storm surges (Figure 5b), $HIDRA_{Ai}$ outperforms $HIDRA_{SLi}$, confirming that atmospheric data is essential for accurate storm surge prediction. Both models achieve similar performance on small residual values, and both perform worse than the full model.

### 4.1.3 Influence of Sea Level Input Type

HIDRA considers the total sea level information split into the tide and the residual provided as separate input time series, and predicts the residual which is added to the tidal signal to predict the full surface height. To analyze the contribution of different sea level input types, two additional variants were considered: (i) $HIDRA_{res}$ considered only the residual as the input to predict the future residuals and (ii) $HIDRA_{sl}$ considered a single total sea level input and predicted the total sea level output.

Results are shown in Table 3. Sea-level-only model ($HIDRA_{sl}$) performs significantly worse than reference $HIDRA_0$ with a 35% increase in RMSE, which speaks in favor of residual prediction over the total sea level. The residuals-only model $HIDRA_{res}$ also performs slightly worse than the full model, causing an 8% RMSE increase, indicating that tidal information as an additional input provides useful context for improved prediction accuracy.

Performance analysis over different prediction lead times (Figure 6a) shows that the sea-level-only model $HIDRA_{sl}$ makes much larger errors (41% increase compared to $HIDRA_0$ ) when predicting far into the future (prediction lead time is high), which suggests that the network has trouble predicting the tidal component that far into the future using only the data from the last 24 hours. Comparing predictions over different residual values (Figure 6b) shows a similar situation. $HIDRA_{sl}$ performs substantially worse (40-50% larger RMSE than $HIDRA_0$ ) for small residual values. This is the range at which the tidal model typically is most accurate and thus provides sufficient information for accurate predictions. Note also that although the sea level only model $HIDRA_{sl}$ does not use tidal information, its prediction errors still follow a similar pattern of increasing

**Table 3.** Performance evaluation of different HIDRA sea level input variants. Results are reported on the entire test set (2014-2016) and separately on storm surge events.

|  | MAE [cm] | RMSE [cm] | Bias [cm] | Likelihood |
|---|---|---|---|---|
| **Overall** | | | | |
| HIDRA$_0$ | **4.9** | **6.4** | -0.4 | **0.0470** |
| HIDRA$_{res}$ | 5.3 | 7.0 | 0.8 | 0.0443 |
| HIDRA$_{sl}$ | 6.6 | 8.7 | **0.2** | 0.0312 |
| Reference (tide) | 12.1 | 15.7 | -2.4 | – |
| **Storm surge events** | | | | |
| HIDRA$_0$ | **10.3** | **12.9** | **-9.3** | **0.0253** |
| HIDRA$_{res}$ | 11.9 | 14.4 | -11.1 | 0.0221 |
| HIDRA$_{sl}$ | 11.7 | 14.4 | -9.6 | 0.0216 |
| Reference (tide) | 49.6 | 50.4 | -49.6 | – |

with growing residual values. This is an interesting result and shows that the examples belonging to small residual values are
inherently easier to predict regardless of whether the tidal estimation is used or not.

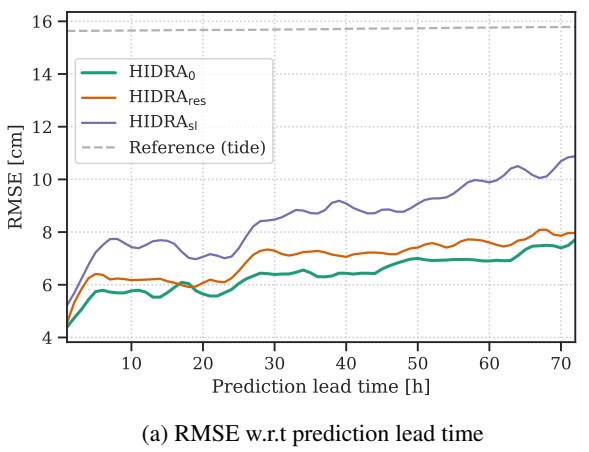

(a) RMSE w.r.t prediction lead time

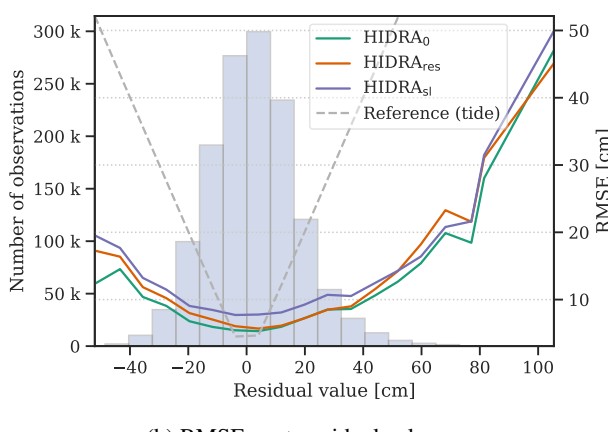

(b) RMSE w.r.t. residual value

**Figure 6.** Performance comparison of HIDRA sea level input variants. HIDRA$_0$ is compared with residuals only model HIDRA$_{res}$, which does not use the estimated tide signal at the input and the sea level only model HIDRA$_{sl}$, which does not use separate tidal estimation and predicts the entire sea level signal. The RMSE with respect to (a) prediction lead time and (b) residual value bins is reported. The errors of the astronomic tide are presented for reference.

### 4.1.4   Influence of the Atmospheric Encoder

The role of trainable discriminative atmospheric encoder ASE is analyzed by replacing it with a reconstructive embedding proposed in Hieronymus et al. (2019). A principal component analysis is applied to the atmospheric input to compute a low-

**Table 4.** Comparison of the proposed ASE-based $HIDRA_0$ and the EOF-based $HIDRA_{EOF}$ in terms of overall performance and performance on storm surges. The astronomic tide model performance is reported for reference.

| | MAE [cm] | RMSE [cm] | Bias [cm] | Likelihood |
|---|---|---|---|---|
| **Overall** | | | | |
| $HIDRA_0$ | **4.9** | **6.4** | -0.4 | **0.0470** |
| $HIDRA_{EOF}$ | 5.0 | 6.6 | **-0.1** | 0.0431 |
| Reference (tide) | 12.1 | 15.7 | -2.4 | – |
| **Storm surge events** | | | | |
| $HIDRA_0$ | **10.3** | **12.9** | **-9.3** | **0.0253** |
| $HIDRA_{EOF}$ | 11.1 | 13.6 | -10.1 | 0.0225 |
| Reference (tide) | 49.6 | 50.4 | -49.6 | – |

dimensional subspace (empirical orthogonal functions, EOFs) that maximizes the data reconstruction. Following Hieronymus
et al. (2019), the top three EOF are used in the subspace construction. The input is projected into this subspace producing a
low-dimensional signal that is directly used in the HIDRA regression network. The modified HIDRA is denoted by $HIDRA_{EOF}$
in the following.

The HIDRA variants with ASE and with EOF are compared in Table 4. In normal conditions, the EOF-based version
($HIDRA_{EOF}$) performs on par with HIDRA using the proposed ASE. The $HIDRA_{EOF}$ RMSE is approximately 3% larger than
that of $HIDRA_0$ . However, the difference increases on the less frequent conditions with high residuals (*i.e.* surges) in which
the EOF-based version results in a 5% RMSE increase compared to ASE-based version. This supports the choice of using
end-to-end learned feature encoder as opposed to a hand-crafted one.

### 4.1.5 Influence of Temporal Encoders

Temporal encoder with temporal attention weights (Section 3.1.2) plays an important part in HIDRA. Two additional variants
are created to study alternative choices of feature encoding. The first HIDRA variant ($HIDRA_{TCN}$) uses temporal convolutional
networks (TCN) (Bai et al., 2018) for encoding the atmospheric and the sea level branch of the network. The atmospheric
branch applies three TCN blocks with 128 units, while the sea level branch applies three TCN blocks with 64 units. The second
HIDRA variant ($HIDRA_{LSTM}$) uses a popular long short-term memory (LSTM) (Hochreiter and Schmidhuber, 1997) networks
for temporal encoding. Three LSTM layers are used in both the atmospheric and the sea level branch of the model. Each layer
contains 128 units in the atmospheric and 64 in the sea level branch.

Results are reported in Table 5 and Figure 7. Overall, $HIDRA_0$ performs on par with the more complex TCN-based
$HIDRA_{TCN}$ (RMSE of $HIDRA_{TCN}$ is 3% larger), while LSTM-based $HIDRA_{LSTM}$ performs worse (RMSE is 14% larger
than $HIDRA_0$ ). $HIDRA_0$ outperforms both TCN and LSTM-based versions by a solid margin on the storm surge events (31%
and 32% RMSE increase, respectively). Furthermore, temporal weights of $HIDRA_0$ use very few parameters compared with
the other variants (see Table 6) – TCN and LSTM-based variations increase the total model size (including other network
layers) by 50% and 150% respectively.

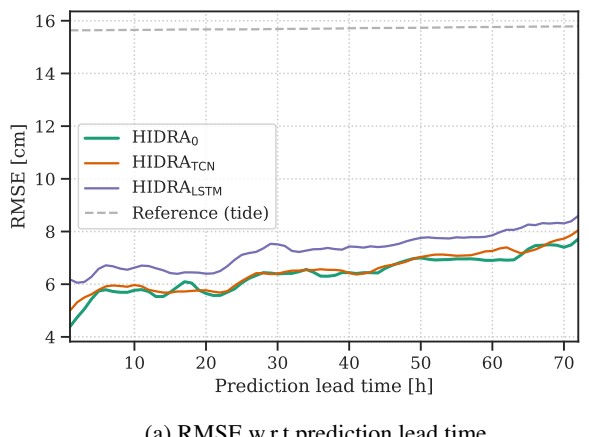 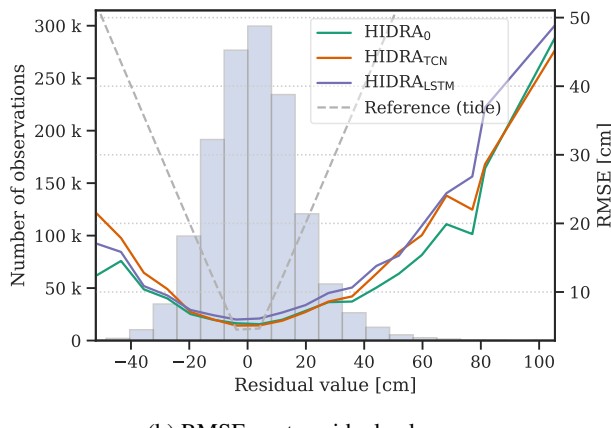

| (a) RMSE w.r.t prediction lead time | (b) RMSE w.r.t. residual value |

**Figure 7.** Comparison of different temporal encoder variants. $HIDRA_0$ is compared with more complex variants using different temporal encoders: LSTMs ($HIDRA_{LSTM}$) and TCNs ($HIDRA_{TCN}$). The RMSE with respect to (a) prediction lead time and (b) residual values is shown. The prediction errors using the astronomic tide are presented for reference.

**Table 5.** Performance of HIDRA model variants using different temporal encoders. $HIDRA_0$ model uses fixed temporal attention for the atmospheric data. $HIDRA_{TCN}$ uses TCNs and $HIDRA_{LSTM}$ uses LSTMs to encode temporal features of atmospheric and sea level data. Performance of the astronomic tide is provided for reference.

|  | MAE [cm] | RMSE [cm] | Bias [cm] | Likelihood |
|---|---|---|---|---|
| **Overall** | | | | |
| $HIDRA_0$ | **4.9** | **6.4** | -0.4 | **0.0470** |
| $HIDRA_{TCN}$ | **4.9** | 6.6 | **-0.2** | 0.0431 |
| $HIDRA_{LSTM}$ | 5.5 | 7.3 | -0.8 | 0.0461 |
| Reference (tide) | 12.1 | 15.7 | -2.4 | - |
| **Storm surge events** | | | | |
| $HIDRA_0$ | **10.3** | **12.9** | **-9.3** | **0.0253** |
| $HIDRA_{TCN}$ | 13.5 | 15.3 | -13.2 | 0.0180 |
| $HIDRA_{LSTM}$ | 13.6 | 16.2 | -12.9 | 0.0188 |
| Reference (tide) | 49.6 | 50.4 | -49.6 | - |

**Table 6.** Total number (in millions) of trainable parameters of HIDRA variants with different temporal encoders.

| Method | # of parameters |
|---|---|
| $HIDRA_0$ | **0.8 M** |
| $HIDRA_{TCN}$ | 1.2 M |
| $HIDRA_{LSTM}$ | 2.1 M |

**Table 7.** Performance of HIDRA variants with different wind and pressure input configurations. $HIDRA_0$ uses the wind stress, $HIDRA_{wnd}$ uses the raw wind, $HIDRA_{no\_wnd}$ does not use wind inputs and $HIDRA_{no\_prs}$ uses the wind stress, but not the air pressure. Performance of the astronomic tide is provided for reference.

| | MAE [cm] | RMSE [cm] | Bias [cm] | Likelihood |
|---|---|---|---|---|
| **Overall** | | | | |
| $HIDRA_0$ | **4.9** | **6.4** | -0.4 | **0.0470** |
| $HIDRA_{wnd}$ | **4.9** | **6.4** | **0.2** | 0.0465 |
| $HIDRA_{no\_wnd}$ | 5.3 | 7.1 | -0.3 | 0.0434 |
| $HIDRA_{no\_prs}$ | 5.3 | 7.0 | -0.1 | 0.0451 |
| Reference (tide) | 12.1 | 15.7 | -2.4 | - |
| **Storm surge events** | | | | |
| $HIDRA_0$ | 10.3 | 12.9 | -9.3 | 0.0253 |
| $HIDRA_{wnd}$ | **9.3** | **11.6** | **-7.8** | **0.0274** |
| $HIDRA_{no\_wnd}$ | 13.7 | 16.5 | -12.9 | 0.0192 |
| $HIDRA_{no\_prs}$ | 11.6 | 14.0 | -10.8 | 0.0225 |
| Reference (tide) | 49.6 | 50.4 | -49.6 | - |

### 4.1.6 Influence of the Wind Input Type and Wind-Pressure Redundancy

Bora and Scirocco characteristics in the Adriatic basin are often determined through an interplay of geostrophic, orographic and other influences (Pasarić et al., 2007; Grisogono and Belušić, 2009). At other times however, non-geostrophic effects may
play a lesser role and the wind field is largely determined by the pressure field. To investigate potential information redundancy between the wind and pressure inputs, two HIDRA variants were trained: one which did not use the wind input and another which used the wind, but not the pressure. Results in Table 7 show that removing either wind or air pressure input leads to an approximately 9% increase of RMSE. HIDRA seems to compensate for potential redundancy in the inputs and capitalizes on the fact that wind in the basin is, in the last instance, not entirely pressure driven. In any case using both inputs is preferred.
We proceed to inspect the impact of Large and Pond parametrization (Large and Pond, 1981) which might oversimplify the wind stress dependence on the wind. To this end we consider another variant of HIDRA, which uses raw wind instead of wind stress. Results in Table 7 show that, overall, the performance between the two wind-input variants is indistinguishable. However, on storm surges, using raw wind reduces the RMSE by approximately 1 cm when compared to the setup which uses wind stress. It appears that HIDRA is capable of extracting the information important for sea level prediction during storm
surges also directly from the raw wind.

### 4.2 Comparisons between HIDRA and NEMO

In regional ocean modeling setups, tides are often implemented as open boundary conditions and are treated as an external part of model's sea level response. In HIDRA on the other hand, tides enter the model as information that gets inextricably linked into its residual (*i.e.* non-tidal part of) prediction: results of Section 4.1.3 show that including tides improves the prediction of
365 the residual itself. Thus in HIDRA, the quantity we refer to as the residual, is in fact composed of two entangled parts: the

**Table 8.** Comparison of HIDRA and NEMO on the Koper 2019 test dataset. Overall performance is shown separately from performance during storm surge events. Performance of the astronomic tide is provided for reference.

| | MAE [cm] | RMSE [cm] | Bias [cm] | Likelihood |
|---|---|---|---|---|
| **Overall** | | | | |
| HIDRA | **8.4** | **10.8** | **0.2** | 0.0323 |
| NEMO | 9.5 | 12.7 | -3.0 | **0.0337** |
| Reference (tide) | 16.0 | 21.0 | -4.7 | - |
| **Storm surge events** | | | | |
| HIDRA | **15.6** | **20.2** | -13.5 | 0.0180 |
| NEMO | 17.1 | 22.4 | **-9.7** | **0.0202** |
| Reference (tide) | 54.9 | 56.7 | -54.9 | - |

atmospheric part of the residual and the error correction of the tidal model. We nevertheless use the term residual to differentiate it not from the tide, but rather from total sea level. For these reasons we cannot, as is otherwise customary in sea level modeling, focus on the verification of the residual part of the total sea level signal. In this section, we thus compare the best-performing variant of HIDRA from the ablation study (Section 4.1.6) and NEMO on the total sea levels, which, in HIDRA and in NEMO, contain both residuals and tides.

### 4.2.1 Overall Performance

HIDRA and NEMO are compared on the 2019 tide-gauge sea level observations in Koper. While HIDRA enables prediction starting at each time-step, NEMO does not, since it runs once per day. The models are thus compared only on the prediction windows matching the NEMO runs. The analysis uses the ensemble mean of NEMO members with a 12 h bias correction, i.e., $\overline{y}_{\mathrm{bc12h}}(t)$ defined in (5). The corresponding ensemble mean HIDRA time series are denoted as $\overline{y}_H(t)$ and the Koper tide gauge time series are denoted by $y^{\mathrm{kp}}(t)$.

Results are reported in Table 8. Overall (top panel of Table 8), HIDRA outperforms NEMO, obtaining a lower MAE, RMSE and bias. During storm surges (bottom panel of Table 8) HIDRA also outperforms NEMO, but exhibits a larger bias. The results visualised with respect to the residual sea level bin in Figure 9 confirm this. To track the prediction error growth with prediction horizon, we show RMSE with respect to the prediciton lead time in Figure 8. Both NEMO and HIDRA exhibit a similar RMSE growth trend of approximately +4 cm per 72 h. But NEMO exhibits a higher mean error and a higher error variance. HIDRA outperforms NEMO over the entire range of prediction lead times with lower and less volatile errors.

To gain further insights, the NEMO, HIDRA and tide gauge 2019 time-series power spectra were analyzed by computing spectral energy densities of the signals over the frequency domain $(2\,\mathrm{h})^{-1} - (96\,\mathrm{h})^{-1}$ (Figure 10). The power spectra were computed as absolute values of one-dimensional Fast Fourier Transforms. The tide gauge power spectrum exhibits clear tidal presence and also a clear peak at the fundamental Adriatic seiche period of 21.5 h. Some higher harmonics are also present in the tide gauge spectrum at shorter periods (below 8 h) which are present in NEMO but absent from HIDRA. They are, however, less important as they contain at least an order of magnitude less energy than tides or the ground state seiche, which may be the

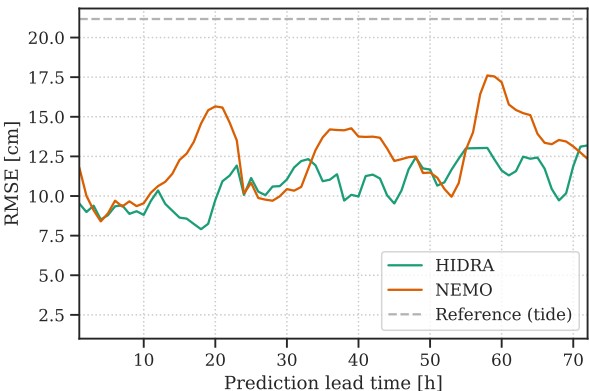

**Figure 8.** Performance comparison of HIDRA and NEMO with respect to the prediction lead time. Error of the astronomic tide is presented for reference as a grey dashed line.

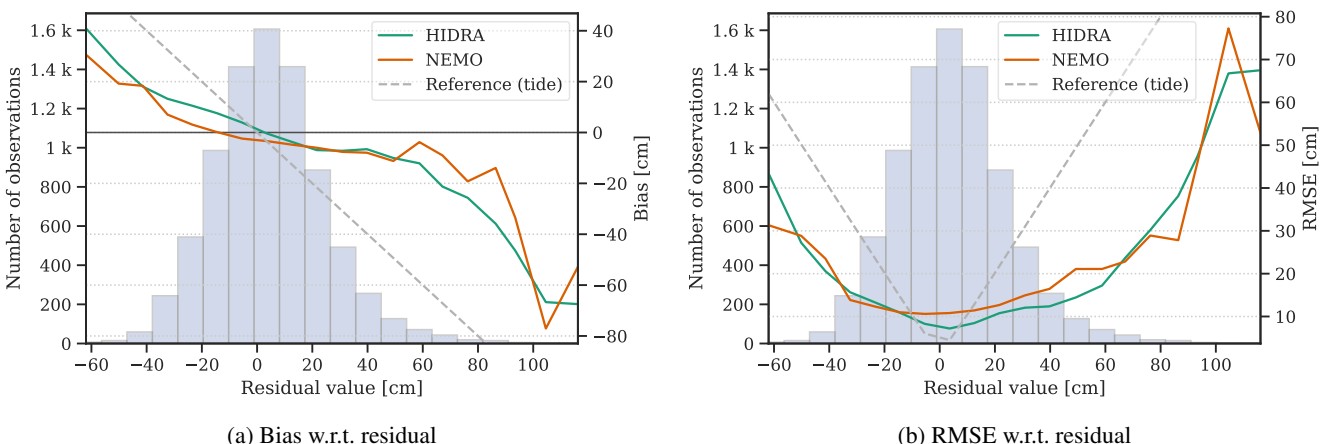

(a) Bias w.r.t. residual

(b) RMSE w.r.t. residual

**Figure 9.** Comparison of HIDRA and NEMO over the sea level distribution. Bias (a) and RMSE (b) with respect to the residual value are shown. Error of the astronomic tide is shown for reference.

reason HIDRA learned to partially ignore it. Both NEMO and HIDRA contain adequate amount of energy at tidal periods. But
NEMO significantly underestimates the amount of energy contained in the frequency band around the fundamental Adriatic seiche. HIDRA, on the other hand, contains an adequate, if slightly underestimated, amount of energy in the seiche frequency band. This seems to be a solid argument to claim that the network has learned to mimic the fundamental basin seiche behaviour. However, adequate HIDRA energy content in the $(21.5\text{h})^{-1}$ frequency band does not in itself mean that Adriatic seiches are excited at appropriate times during storm surge events. To test whether this is indeed the case, inspection of specific storm
surge cases is required.

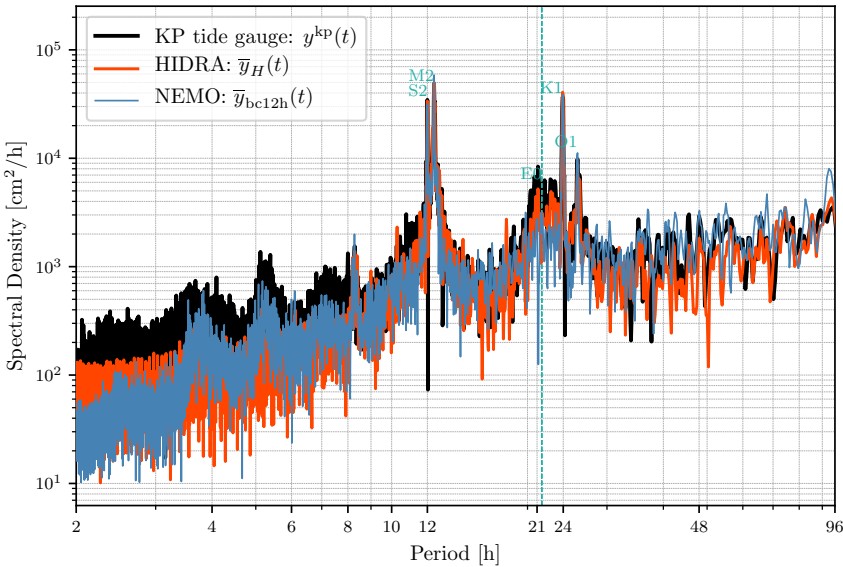

**Figure 10.** Power spectrum of NEMO $\overline{y}_{\text{bc12h}}(t)$ ensemble forecast mean time-series (blue line), HIDRA ensemble forecast mean time-series $\overline{y}_H(t)$ (orange line) and tide gauge observations time-series $y^{\text{kp}}(t)$ (black line) for year 2019. Turquoise symbols denote spectral peaks due to respective tidal constituent. $E0$ denotes the spectral peak due to fundamental Adriatic seiche with a period of 21.5 h, also marked with the vertical turquoise dotted line.

### 4.2.2   Specific Storm Surge Events

We now proceed to investigate the total sea level time-series predicted by NEMO and HIDRA during specific storm surges by analyzing the total sea level ensemble envelopes from a 12-hour bias corrected NEMO $\overline{y}_{\text{bc12h}}(t)$ and HIDRA $\overline{y}_H(t)$. In addition, continuous wavelet transforms (CWT, see *e.g.* Mallat (2009)) over time windows containing the specific storm surges
are computed. This allows comparison of the excitation level of specific harmonic contributions to the total sea level during each particular storm surge event. The analysis is focused on the semi-diurnal tidal signal (with periods around 12 h) and on the fundamental seiche period (with a period of 21.5 h). A Morlet wavelet was used for the CWT convolution computation.

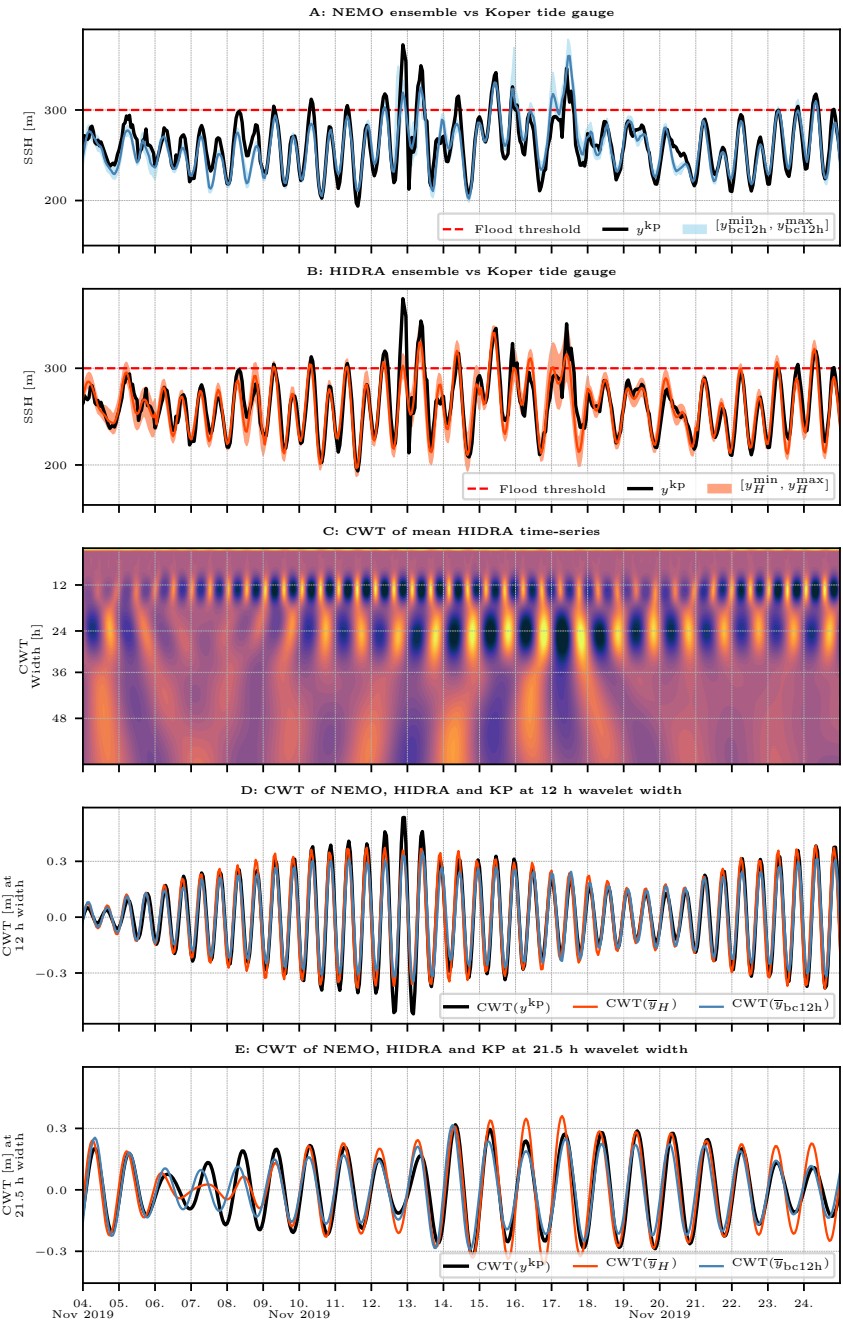

**Figure 11.** Comparison between NEMO (with a 12-hour bias correction) and HIDRA ensembles during a historic surge of November 2019. Panels A and B depict NEMO and HIDRA ensemble predictions for Koper sea level, together with Koper tide gauge time series. Mean values and minimum-maximum ensemble spans are depicted, as defined in Section 3.3. Panel C depits a CWT of $\overline{y}_H(t)$ HIDRA mean timeseries. Panels D and E depict a CWTs of Koper tide gauge, HIDRA and NEMO time-series at semi-diurnal (12 h) and E0 (21.5 h) wavelet width respectively. Flooding threshold in the historic coastal town of Piran (Slovenia) is marked by a dashed red line in the top two panels.

We first discuss the historic storm surge flooding event from mid-November 2019. Atmospheric conditions during November 2019 were not remarkable in themselves. Mean sea level pressures were moving between 990-1000 mbar, while Scirocco speeds in Northern Adriatic were measured to be around 10 ms$^{-1}$. However, an unfortunate coinciding of a general low pressure, high neap tide and seiche-inducing high-frequency forcing due to a local pressure low caused one of the worst Northern Adriatic floods in history (Cavaleri et al., 2020). These multifaceted circumstances made forecasting of these floods, using a lower resolution forcing such as ECMWF ensemble, challenging (Cavaleri et al., 2020). This problem is at least partly reflected in the November 2019 forecasts, presented in Figure 11.

Panel A in Figure 11 depicts NEMO $\overline{y}_{bc12h}(t)$ mean time-series, together with the $y_{bc12h}^{min}(t) - y_{bc12h}^{max}(t)$ ensemble envelope, while panel B depicts HIDRA predictions. Both NEMO and HIDRA mean time-series seem to underestimate the storm surge peak values. Both ensembles, however, exhibit high forecast spread, with maximums often adequately representing the observed peaks. The first peak on 12th November 2019 is missed by HIDRA, but the subsequent dynamics is better represented in HIDRA than in NEMO. In particular, HIDRA does not exhibit a substantial false positive on 15th November and also overshoots less during the surge of 17th November 2019. Judging from CWT signals of mean ensembles from both HIDRA and NEMO (in panels D and E), HIDRA missing the first peak can be at least partly attributed to underestimation of semi-diurnal tidal signal in the HIDRA forecast. Semi-diurnal tides are similarly represented in both models and both underestimate the signal in this band. On the other hand the seiche signal seems better represented in HIDRA (panel E) during the storm surge. Note that HIDRA excites the seiche immediately after the sea level peak on 12th November. NEMO, of course, cannot do this since the seiche period is 21.5 h. Panel D of Figure 11 show that, like NEMO, HIDRA resolves well the low frequency tidal variability between spring and neap tides.

We now move to an event from late January and early February 2015, which turned out to be quite problematic for NEMO to forecast, while HIDRA behaved much better. During this period, Adriatic was impacted by several days of low pressures (990-1000 mbar) and moderate Scirocco (with speeds 8-12 ms$^{-1}$). These conditions led to a series of moderate storm surges in the Northern Adriatic, as shown in Figure 12.

NEMO ensemble, depicted in Panel A of Figure 12, performs particularly poorly during this time window. While it did predict the first surge on 30th January, the following peaks were underestimated and the crest-to-trough sea level range of NEMO is overall unsatisfactory throughout the time-window. Since the tidal part of the NEMO signal is appropriate (Figure 12, Panel D), the reason for poor forecast seems to lie in insufficient excitation of the fundamental basin seiche, which is drastically underestimated in our setup of NEMO (Figure 12, Panel E). HIDRA yields a more accurate overall forecast in this case (Figure 12, panel B), and does not underestimate the seiche signal at the $(21.5 \text{ h})^{-1}$ frequency as much as NEMO. Semi-diurnal tidal signal is reasonably well represented in both models (panel D of Figure 12). These performances of NEMO and HIDRA are consistent with the power spectrum in Figure 10.

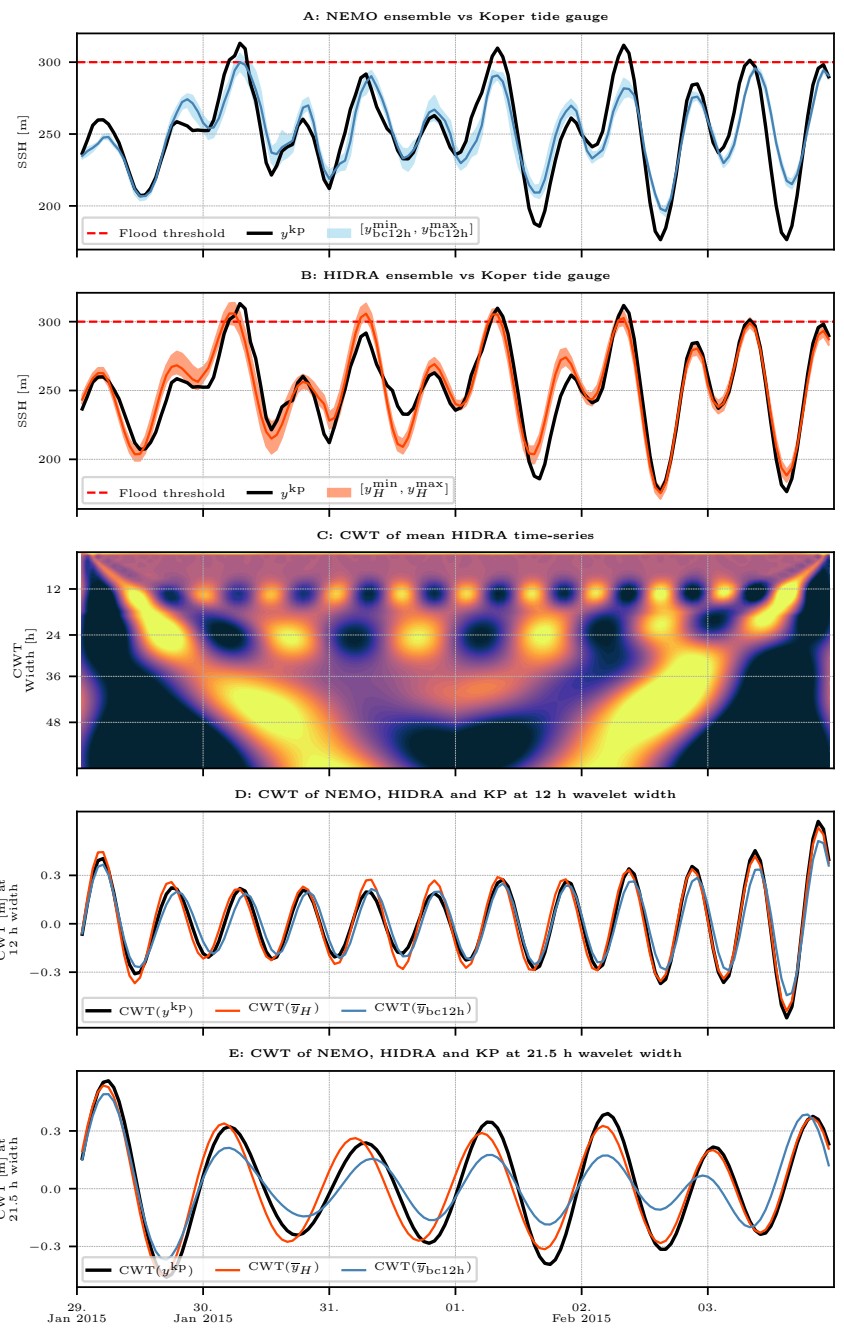

**Figure 12.** Comparison between NEMO (with a 12-hour bias correction) and HIDRA ensembles during surge of January and February 2015. Panels A and B depict NEMO and HIDRA ensemble predictions for Koper sea level, together with Koper tide gauge time series. Mean values and minimum-maximum ensemble spans are depicted, as defined in Section 3.3. Panel C depits a CWT of $\overline{y}_H(t)$ HIDRA mean timeseries. Panels D and E depict a CWTs of Koper tide gauge, HIDRA and NEMO time-series at semi-diurnal (12 h) and E0 (21.5 h) wavelet width respectively. Severe flooding threshold in the historic coastal town of Piran (Slovenia) is marked by a dashed red line in the top two panels.

## 5 Conclusions

In this study, we presented HIDRA, a novel deep learning network for sea level modeling in complex environments like the Adriatic. We describe key HIDRA architecture blocks and discuss several aspects of how both HIDRA architecture and its input influence its performance. HIDRA compares favorably to the current operational NEMO setup of the National Hydrological Forecasting Service at Slovenian Environment Agency. While further tuning of the operational NEMO setup at the Agency is also under way (with the aim of improving its forecasting skill), results presented in this study nevertheless indicate that HIDRA is an appropriate candidate for the Agency's operational pipeline. Preliminary tests (not reported in this study) indicate that HIDRA also generalizes well to other geographical locations.

Last but not least, numerical cost of both setups is vastly different. NEMO ensemble runs require dedicated HPC facilities, while the HIDRA ensemble forecast can be executed on a personal computer (even without a dedicated GPU) and exhibits an extremely low energy footprint. A single HIDRA run for our requirements takes less than half a CPU second per ensemble member, while a full basin NEMO ensemble requires tens of CPU hours per ensemble member – a speedup in order of $0.5 \times 10^6$ times.

We believe the presented results are a promising first step. In our future work we plan to focus on improving the performance of both HIDRA and NEMO in the tails of the sea level distributions as well as explore other environmental input streams and architectural designs to further reduce the prediction errors with increasing forecast horizon. We hope this study builds a strong case in favor of machine learning capabilities with carefully designed architectures to discern sea level dynamics in regional basins and will inspire other groups to consider similar solutions.

*Code and data availability.* HIDRA code and data samples are available in the Git repository: https://github.com/lojzezust/HIDRA (last access: 5 January 2021). Persistent version of the HIDRA 1.0 source code is available through https://doi.org/10.5281/zenodo.4457305 (Žust et al., 2020a). ECMWF ensemble data are available through the Meteorological Archive and Retrieval System (MARS), but access is limited to member countries. Sea level datasets employed in this paper are available at https://doi.org/10.5281/zenodo.4106440 (Žust et al., 2020b) and NEMO configuration namelist used in the experiments is published at https://doi.org/10.5281/zenodo.4419333 (Žust et al., 2021).

*Author contributions.* LŽ and MK designed and implemented HIDRA. ML provided the physics-related background for HIDRA architecture. AF obtained and preprocessed ECMWF ensemble data. AF and ML implemented NEMO ensemble. ML, LŽ and MK analyzed the results and wrote the paper. All authors contributed to the final version of the manuscript.

*Competing interests.* Authors declare no competing interests.

*Acknowledgements.* ML wishes to acknowledge financial support of the Slovenian Research Agency (ARRS) project grant J1-9157: "Drivers that structure coastal marine microbiome with emphasis on pathogens – an integrated approach". MK and LŽ were supported in part by the the Slovenian Research Agency (ARRS) basic research project grant J2-2506 "Adaptive deep perception methods for autonomous surface vehicles" and research program P2-0214 "Computer vision". The paper benefited from comments of two anonymous reviewers.

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
