# Peer review of "HIDRA 1.0: Deep-Learning-Based Ensemble Sea Level Forecasting in the Northern Adriatic"

_Geoscientific Model Development, 2020_

## Short Comment (SC1) · 14 Nov 2020

Dear authors,

in my role as Executive editor of GMD, I would like to bring to your attention our Editorial version 1.2:

https://www.geosci-model-dev.net/12/2215/2019/

This highlights some requirements of papers published in GMD, which is also available on the GMD website in the 'Manuscript Types' section: http://www.geoscientific-model-development.net/submission/manuscript_types.html

[Figure]

In particular, please note that for your paper, the following requirement has not been met in the Discussions paper:

- Code must be published on a persistent public archive with a unique identifier for the exact model version described in the paper or uploaded to the supplement, unless this is impossible for reasons beyond the control of authors. All papers must include a section, at the end of the paper, entitled "Code availability". Here, either instructions for obtaining the code, or the reasons why the code is not available should be clearly stated. It is preferred for the code to be uploaded as a supplement or to be made available at a data repository with an associated DOI (digital object identifier) for the exact model version described in the paper. Alternatively, for established models, there may be an existing means of accessing the code through a particular system. In this case, there must exist a means of permanently accessing the precise model version described in the paper. In some cases, authors may prefer to put models on their own website, or to act as a point of contact for obtaining the code. Given the impermanence of websites and email addresses, this is not encouraged, and authors should consider improving the availability with a more permanent arrangement. Making code available through personal websites or via email contact to the authors is not sufficient. After the paper is accepted the model archive should be updated to include a link to the GMD paper.

As GitHub is not a persistent archive, please provide a persistent release for the exact source code version used for the publication in this paper. As explained in https://www.geoscientific-model-development.net/about/manuscript_types.html the preferred reference to this release is through the use of a DOI which then can be cited in the paper. For projects in GitHub a DOI for a released code version can easily be created using Zenodo, see https://guides.github.com/activities/citable-code/ for details.

Yours, Astrid Kerkweg

---

## Referee Comment (RC1) · Anonymous Referee #1 · 16 Nov 2020

This manuscript designed a deep-learning-based model to predict the sea level in the Northern Adriatic. It comprehensively analyzed the model performances under different model structures and then compared it with a numerical model (NEMO v3.6). It is interesting and worth being published after clarifying some minor issues:

1) The authors tend to apply the HIDRA model for business forecasting. However, the input for the model contains the atmospheric data in the future. Please clarify the data source in the business forecast process.

2) The authors declare "Extending the historical horizon beyond 24 hours did not significantly affect the prediction accuracy". Please give a concise description of how to

find the trade-off between the model forecast accuracy and the computing resource.

3) In Equation (1), there is a "20" on the "sum" signal, which represents the different spatial position on the feature maps, which is confusing that where this value comes from. Please clarify the changes of the feature maps during the fore-propagation process in the Figure. 3, especially in Figure 3(a). Such as marking the size of the convolution kernel and the output size in the red boxes.

4) Line 298: results → result.

---

## Author Comment (AC1) · 16 Nov 2020

Dear Astrid Kerkweg,

thank you very much for bringing this to our attention.

We have prepared a persistent release of the source code. It is available through the DOI: 10.5281/zenodo.4274708.

How do you suggest we update this information? As far as I can see, there is currently no option in the submission portal to update the manuscript records or the manuscript itself.

[Figure]

Kind regards,

Lojze Žust

---

## Referee Comment (RC2) · Anonymous Referee #2 · 20 Nov 2020

This manuscript proposed a model named HIDRA based on a deep learning network to forecast the sea level in the Adriatic. This is a good try and shows the deep-learning-based model has a good future in ocean environment research and forecasting. This work is worthwhile and the manuscript is well-written in general. However, there are still several critical issues to be clarified.

Major comments:

1) The Large and Pond parameterization is suitable for calculating the wind stress over the open sea with deep water. In other words, this scheme cannot use in this study. As the data is the basic and core of machine learning, the authors should find another

scheme to redo this work.

2) In fact, the wind is associated with sea level pressure and latitude (Coriolis force). For the Adriatic, the difference of wind in different locations due to Coriolis force can be ignored, which means the wind is almost determined by sea level pressure. So, wind stress has included information on sea level pressure. I think it's double-counted when the authors used wind stress as well as sea level pressure.

3) Topography is important for the sea level, besides the wind stress (or sea level pressure). Therefore, topography should also be considered in the HIDRA model.

4) Line 327, the description is inaccurate. Usually, the regional ocean model, especially the coastal model, can simulate the sea level including the tidal directly by using the water level boundary condition. Moreover, the authors cannot claim HIDRA is better than NEMO because they only compared with results from only one NEMO configuration they used. If the NEMO is tuned carefully, maybe the results are better. In fact, the NEMO in the storm surge events seems better than HIDRA.

Minor comments:

1) How to deal with the land points in this study is missed.

2) Why did the authors select the 29x37 for the atmospheric tensor?

3) It's better to give a table for the HIDRA and NEMO configuration.

———————————————————

---

## Author Comment (AC2) · 20 Nov 2020

**Authors' response to Anonymous Referee #1**
* * *
We thank the reviewer for their time and comments that helped us to improve the paper. Responses to individual comments are provided below point-by-point. We also paste the text to reflect the changes to the manuscript.
* * *
**Comment 1:** The authors tend to apply the HIDRA model for business forecasting. However, the input for the model contains the atmospheric data in the future. Please clarify the datasource in the business forecast process.

**Response:** Thank you for pointing this out. As is usual in supervised learning, HIDRA was trained on past atmospheric model forecasts and sea level data. The HIDRA training data is composed of two parts: (a) the atmospheric part consists of past forecasts from a single member of ECMWF model ensemble, and (b) the sea level training data consists of tidal and residual sea levels from 24 hours prior to the forecast obtained from observations in Koper. For operational forecast, starting at time $t_0$, the following datasets will be employed:

- The atmospheric data: ECMWF ensemble forecast (the same ECMWF product as in training data) for time interval $[t_0, t_0 + 72$ hours$]$. This data is available at the time of the forecast from ECMWF operational service.

- Sea Level data: tidal and residual data for the time interval $[t_0 - 24$ hours, $t_0]$. This data is also available at the time of the forecast, assuming that the tide gauge in Koper works as expected, which is a reasonable assumption, given the redundancy in the design (the gauge consists of three independent bottom-mounted pressure gauges).

All the data required for HIDRA forecasting will be available for operational forecasts every day. To make this clear, the following text in the Conclusion,

*"HIDRA outperforms the current operational NEMO setup and is therefore an appropriate candidate for Agency's operational pipeline."*

was re-written into:

*"HIDRA outperforms the current operational NEMO setup and is therefore an appropriate candidate for Slovenian Environment Agency's operational pipeline. HIDRA integration should be straightforward since ECMWF ensemble predictions and tide gauge sea level data are available at the Agency every day in real time for operational forecasting."*

**Comment 2:** The authors declare "Extending the historical horizon beyond 24 hours did not significantly affect the prediction accuracy". Please give a concise description of how to find the trade-off between the model forecast accuracy and the computing resource.

**Response:** To improve the insight into the trade-off, we conducted an additional experiment in which HIDRA was re-trained and tested for different values of historic horizon $T_{\min}$. The results are shown in Table 1. We find that increasing the historic horizon beyond 24h does not yield measurable improvements in prediction accuracy while increasing the number of parameters in the lower layers and negatively impacting the computational performance. For this reason we decided to set $T_{\min} = 23$.

|  | MAE [cm] | RMSE [cm] | Bias [cm] | Likelihood | CPU time [s] |
|---|---|---|---|---|---|
| **Overall** | | | | | |
| $\text{HIDRA}_{12}$ | 5.3 | 7.0 | **-0.2** | 0.0440 | **0.17** |
| $\text{HIDRA}_{24}$ | 4.9 | **6.4** | -0.4 | **0.0470** | 0.19 |
| $\text{HIDRA}_{36}$ | **4.8** | **6.4** | -0.9 | 0.0455 | 0.21 |
| $\text{HIDRA}_{48}$ | **4.8** | **6.4** | -0.6 | 0.0438 | 0.23 |
| **Storm surge events** | | | | | |
| $\text{HIDRA}_{12}$ | 11.7 | 13.9 | -11.2 | 0.0220 | **0.17** |
| $\text{HIDRA}_{24}$ | **10.3** | 12.9 | -9.3 | **0.0253** | 0.19 |
| $\text{HIDRA}_{36}$ | 10.7 | 13.2 | -9.8 | 0.0245 | 0.21 |
| $\text{HIDRA}_{48}$ | 10.4 | **12.8** | **-9.1** | 0.0251 | 0.23 |

Table 1: HIDRA performance for different historic horizons in terms of mean absolute error (MAE), root mean squared error (RMSE) and model bias. CPU execution time (on a single core) per example is also reported. Performance for the atmospheric tide is provided as reference.

This experiment is now described along with the Table 1 in a new section in the manuscript (Section 4.1.1: Influence of the historic horizon):

*"We first analyze the influence of the HIDRA historic horizon defined by the parameter $T_{\min}$ (see Section 3). Table 1 summarizes the performance of HIDRA with $T_{\min} \in \{11, 23, 31, 47\}$, which translates to historic horizons of 12, 24, 36 and 48 hours. Increasing the historic horizon from 12 to 24 hours significantly improves the prediction accuracy (9% reduction in RMSE error), however, further increases of the historic horizon (i.e., to 36 or 48 hours) do not show measurable benefits. Note that the execution time increases with the length of the historic horizon due to a substantial increase of parameters on the input layer. For this reason, we use a historic horizon of 24 hours ($T_{\min} = 23$) as the best trade-off in the remaining analysis."*

**Comment 3:** In Equation (1), there is a "20" on the "sum" signal, which represents the different spatial position on the feature maps, which is confusing that where this value comes from. Please clarify the changes of the feature maps during the fore-propagation process in the Figure. 3, especially in Figure 3 (a). Such as marking the size of the convolution kernel and the output size in the red boxes.

**Response:** Thank you for directing our attention to this issue. We agree that it is not immediately clear that the "20" in the spatial position sum follows from the input feature maps are of size $4 \times 5$. To address this, we now make the indexing in equation (1) explicit. In particular, the equation

$$\mathbf{f}_t = \text{ReLU}\left(\sum_{i=1}^{20} \mathbf{F}_t^{(i)} \mathbf{w}_t^{(i)}\right) \tag{1}$$

is now changed to

$$\mathbf{f}_t = \text{ReLU}\left(\sum_{i=1}^{4}\sum_{j=1}^{5} \mathbf{F}_t^{(i,j)} \mathbf{w}_t^{(i,j)}\right), \tag{2}$$

where $(i, j)$ denotes the spatial coordinates of the feature map and spatial weights. We believe this is more precise and easier to follow.

Next, we address the comments regarding Figure 3a. We updated Figure 3 in the manuscript and made the annotations in the figure (see Figure 1) and the caption consistent with with the text in the paper's body. The Atmospheric Spatial Encoder now contains the standard markings of ResNet stages as well as the output feature map size for context. The previous caption of Figure 3

*"The proposed HIDRA architecture. A convolutional Atmospheric Spatial Encoder (ASE) extracts spatial atmospheric features from each time-step. Atmospheric and sea level temporal features are encoded by respective Temporal Encoder blocks, fused and passed to the fully-connected Residual Regression Block to predict the residuals along with their uncertainties. The trainable blocks are denoted by red color."*

was changed to

*"The proposed HIDRA architecture. A convolutional Atmospheric Spatial Encoder (ASE) extracts spatial atmospheric features from each time-step. Atmospheric and sea level temporal features are encoded by respective Temporal Encoder blocks, fused and passed to the fully-connected Residual Regression Block to predict the residuals along with their uncertainties. With n we denote the number of filters or units of the block. The trainable blocks are colored red. The structure of the bottleneck blocks used in the ASE is presented in Figure 4."*

Furthermore, we present the structure of the Bottleneck blocks in an additional figure (see Figure 2), which also details the convolutional parameters (kernel sizes, stride) and output feature map sizes.

**Comment 4:** Line 298: results $\rightarrow$ result

**Response:** Thank you for for spotting the grammar mistake. The manuscript was updated as suggested.

[Figure]

Figure 1: The proposed HIDRA architecture. A convolutional *Atmospheric Spatial Encoder* (ASE) extracts spatial atmospheric features from each time-step. Atmospheric and sea level temporal features are encoded by respective Temporal Encoder blocks, fused and passed to the fully-connected *Residual Regression Block* to predict the residuals along with their uncertainties. With $n$ we denote the number of filters or units of the block. The trainable blocks are colored red. The structure of the bottleneck blocks used in the ASE is presented in Figure 2.

[Figure]

Figure 2: Structure of the bottleneck blocks used in the Atmospheric Spatial Encoder (Figure 1). The bottleneck block takes a feature map with depth $F_{in}$ as the input and outputs a feature map with depth $F_{out}$. A regular bottleneck block (left) retains the spatial dimensions of the feature maps, while the downsampling (DS) bottleneck block (right) uses strided convolutions to reduce the spatial dimensions in half. We denote the number of convolutional filters by $n$ and the stride parameter by $s$.

---

## Short Comment (SC2) · 23 Nov 2020

Dear Zojze Zust,

as you posted this DOI here in the discussion forum, it is sufficient to update the manuscript upon revision.

Regards,

Astrid Kerkweg

---

## Author Comment (AC3) · 30 Nov 2020

**Authors' response to the Anonymous Referee #2**

We thank the reviewer for their time and comments that helped us to improve the paper. Detailed responses to individual comments are provided below point-by-point. We also paste the text to reflect the changes to the manuscript. In summary, two major changes have been made in line with the reviewer's suggestions:

- We have turned off the wind stress parametrization and repeated training and analyses with wind-only input data. Results with raw wind input is indeed better than with the wind stress transformation. We now use raw wind input during training. The manuscript has been modified accordingly.
- We have performed another sensitivity study to check the information redundancy in sea level pressure / wind input data. The HIDRA model was re-trained with wind-only and pressure-only inputs. The best results are obtained when both inputs are provided to HIDRA indicating complementarity of the two inputs and ability of HIDRA to compensate for any potential redundancy.
* * *
**Comment 1: This manuscript proposed a model named HIDRA based on a deep learning network to forecast the sea level in the Adriatic. This is a good try and shows the deep-learning-based model has a good future in ocean environment research and forecasting. This work is worthwhile and the manuscript is well-written in general.**

**Response:** We thank the reviewer for the encouraging comment.

**Comment 2: However, there are still several critical issues to be clarified. The Large and Pond parameterization is suitable for calculating the wind stress over the open sea with deep water. In other words, this scheme cannot use in this study. As the data is the basic and core of machine learning, the authors should find another scheme to redo this work.**

**Response:** We thank the reviewer for this insight. Indeed, Large and Pond formulation has its limits, but is often applied in oceanography as a tradeoff between complexity and accuracy. In our case it is worth pointing out that this particular parametrization was not chosen to most concisely represent the vertical momentum flux at the sea surface (which would admittedly require more complex schemes and more data), but to merely introduce the nonlinear wind stress dependence on the wind. Thus our reasoning was that this would make learning the relationships between the atmospheric conditions and tide gauge sea level easier for the deep network.

However, following the referee's insight, we have performed a number of new experiments. In particular, HIDRA was re-trained with using the raw wind instead of Large and Pond parametrization. The prediction performance did not change overall, which means that the network is capable of modeling, at some level, how wind translates into vertical momentum flux. We did notice that the

prediction performance on storm surges did improve on average by approximately 1 cm. These results will be added to the revised manuscript as a separate subsection in the "Hidra Architecture Analysis" Section 4.1. For convenience we include the new subsection below:

*"We proceed to inspect the impact of Large and Pond parametrization which might oversimplify the wind stress dependence on the wind. To this end we consider another variant of HIDRA, which uses raw wind instead of wind stress. Results in Table 7 show that, overall, the performance between the two wind-input variants is indistinguishable. However, on storm surges, using raw wind reduces the RMSE by approximately 1 cm when compared to the setup which uses wind stress. It appears that HIDRA is capable of extracting the information important for sea level prediction during storm surges also directly from the raw wind."*

**Table 7.** Performance of HIDRA variants with different wind and pressure input configurations. $HIDRA_0$ uses the wind stress, $HIDRA_{wnd}$ uses the raw wind, $HIDRA_{no\_wnd}$ does not use wind inputs and $HIDRA_{no\_prs}$ uses the wind stress, but not the air pressure. Performance of the astronomic tide is provided for reference.

| | MAE [cm] | RMSE [cm] | Bias [cm] | Likelihood |
|---|---|---|---|---|
| **Overall** | | | | |
| $HIDRA_0$ | **4.9** | **6.4** | -0.4 | **0.0470** |
| $HIDRA_{wnd}$ | **4.9** | **6.4** | 0.2 | 0.0465 |
| $HIDRA_{no\_wnd}$ | 5.3 | 7.1 | -0.3 | 0.0434 |
| $HIDRA_{no\_prs}$ | 5.3 | 7.0 | -0.1 | 0.0451 |
| Reference (tide) | 12.1 | 15.7 | -2.4 | - |
| **Storm surge events** | | | | |
| $HIDRA_0$ | 10.3 | 12.9 | -9.3 | 0.0253 |
| $HIDRA_{wnd}$ | **9.3** | **11.6** | **-7.8** | **0.0274** |
| $HIDRA_{no\_wnd}$ | 13.7 | 16.5 | -12.9 | 0.0192 |
| $HIDRA_{no\_prs}$ | 11.6 | 14.0 | -10.8 | 0.0225 |
| Reference (tide) | 49.6 | 50.4 | -49.6 | - |

**Comment 3: In fact, the wind is associated with sea level pressure and latitude (Coriolis force).For the Adriatic, the difference of wind in different locations due to Coriolis force can be ignored, which means the wind is almost determined by sea level pressure. So, wind stress has included information on sea level pressure. I think it's double-counted when the authors used wind stress as well as sea level pressure.**

**Response:** Thank you for the comment. Indeed, winds in the Adriatic basin have a substantial geostrophic component. But we would like to note that the two strongest winds, Bora and Scirocco, can also have a significant non-geostrophic contribution (orographic channeling, non-linear wave breaking, density-driven flows during weak Bora episodes etc., *e.g.* Pasarić et al., 2007, https://angeo.copernicus.org/articles/25/1263/2007/; Grisogono and Belušić, 2009, https://onlinelibrary.wiley.com/doi/10.1111/j.1600-0870.2008.00369.x). In such cases the wind field can exhibit cross-isobaric flow and reflects other constraints beyond the geostrophic equilibrium.

Thus, to explicitly verify the potential effect of the information redundancy between the wind and air pressure inputs, we trained several additional variants of HIDRA. One that used wind input but not pressure, and another that used the air pressure but not the wind. Results show that performance drops by approximately 10% when excluding either wind or air pressure, implying that the network

accounts for potential redundancy and capitalizes on the complementary information encoded in both inputs. These results will be added in the revised manuscript as a separate subsection in the "Hidra Architecture Analysis" Section 4.1. For convenience we include the new subsection below:

*"Bora and Scirocco characteristics in the Adriatic basin are often determined through an interplay of geostrophic, orographic and other influences (Pasarić et al., 2007; Grisogono and Belušić, 2009). At other times however, non-geostrophic effects may play a lesser role and the wind field is largely determined by the pressure field. To investigate potential information redundancy between the wind and pressure inputs, two HIDRA variants were trained: one which did not use the wind input and another which used the wind, but not the pressure. Results in Table 7 show that removing either wind or air pressure input leads to an approximately 9% increase of RMSE. HIDRA seems to compensate for potential redundancy in the inputs and capitalizes on the fact that wind in the basin is not entirely pressure driven. In any case using both inputs is preferred."*

**Table 7.** Performance of HIDRA variants with different wind and pressure input configurations. $HIDRA_0$ uses the wind stress, $HIDRA_{wnd}$ uses the raw wind, $HIDRA_{no\_wnd}$ does not use wind inputs and $HIDRA_{no\_prs}$ uses the wind stress, but not the air pressure. Performance of the astronomic tide is provided for reference.

|  | MAE [cm] | RMSE [cm] | Bias [cm] | Likelihood |
|---|---|---|---|---|
| **Overall** | | | | |
| $HIDRA_0$ | **4.9** | **6.4** | -0.4 | **0.0470** |
| $HIDRA_{wnd}$ | **4.9** | **6.4** | **0.2** | 0.0465 |
| $HIDRA_{no\_wnd}$ | 5.3 | 7.1 | -0.3 | 0.0434 |
| $HIDRA_{no\_prs}$ | 5.3 | 7.0 | -0.1 | 0.0451 |
| Reference (tide) | 12.1 | 15.7 | -2.4 | - |
| **Storm surge events** | | | | |
| $HIDRA_0$ | 10.3 | 12.9 | -9.3 | 0.0253 |
| $HIDRA_{wnd}$ | **9.3** | **11.6** | **-7.8** | **0.0274** |
| $HIDRA_{no\_wnd}$ | 13.7 | 16.5 | -12.9 | 0.0192 |
| $HIDRA_{no\_prs}$ | 11.6 | 14.0 | -10.8 | 0.0225 |
| Reference (tide) | 49.6 | 50.4 | -49.6 | - |

**Comment 4: Topography is important for the sea level, besides the wind stress (or sea level pressure). Therefore, topography should also be considered in the HIDRA model.**

**Response:** We agree that topography affects the sea level dynamics and is imperative for sea level prediction in physical models. We would nevertheless like to point out that topography is constant and does not change with time, thus it is not necessary to provide it explicitly as an input parameter to HIDRA. Note that HIDRA is not a dynamical model which would have to explicitly take into account the complex per-point topography interaction. Rather, interactions between dynamic spatially-varying and static elements (like topography), relevant for sea level prediction accuracy, are *learned implicitly* by the deep neural network from the vast amount of data. To achieve this in HIDRA, the spatial encoding is enforced by providing the normalized spatial coordinates as part of the input to the network (*x* and *y* input channels), thus allowing the learning algorithm to make such spatially-dependent relations.

Furthermore, HIDRA predicts the sea level for a single specific location, taking into account the past sea level measurements from that location (Koper tide gauge). These measurements already reflect Adriatic basin bathymetry: sea level experiences topographic amplification due to shallowness in the north, resonant amplification due to forcing frequency being close to the basin seiche frequency, and also reflection due to the closure of the basin in the north. All these bathymetry effects are already implicitly contained in the observations that the network receives as the input. In fact, our experiments show that HIDRA can respond in a manner that consistently reflects these bathymetric constraints, for example that it amplifies the signal in the basin seiche $(21.5 \text{ hr})^{-1}$ frequency band during a Scirocco event.

**Comment 5: Line 327, the description is inaccurate. Usually, the regional ocean model, especially the coastal model, can simulate the sea level including the tidal directly by using the water level boundary condition.**

**Response:** Thank you for pointing this out. The description will be updated by including NEMO configuration namelist into the paper supplement (the parameter space of the NEMO model is too large to be completely covered in the descriptive text). We also agree that regional ocean models can simulate sea level (including tides) via open boundary conditions.

As pointed out by the referee, tides are not included in the forcing of the current NEMO setup. This decision is partly based on the fact that we have a tide-gauge in Koper and we can analyze the tidal constituents for Koper directly from the local observations, which seemed to be the most straightforward way of obtaining tides in Koper. We have however in the past compared full (with tides on open boundaries) NEMO sea-levels to the setup presented in this paper. The main result (unpublished) was that sea level from NEMO with tides at the open boundary offers comparable, but somewhat worse representation of observed total sea levels than the non-tidal setup of NEMO with tides computed on the Koper tide gauge.

Furthermore, as explained in the paper, HIDRA is using tidal sea-levels and residuals obtained from the Koper tide-gauge. Allowing NEMO and HIDRA to use the same tidal signal allows for more consistent comparisons of their performance. We have therefore chosen not to include tidal forcing at NEMO open boundary (in the Ionian Sea) in this study, but to obtain the tidal part of the sea level signal from local observations.

**Comment 6: Moreover, the authors cannot claim HIDRA is better than NEMO because they only compared with results from only one NEMO configuration they used. If the NEMO is tuned carefully, maybe the results are better. In fact, the NEMO in the storm surge events seems better than HIDRA.**

**Response:** We thank the reviewer for this comment. We agree -- there certainly exists a possibility that a better tuned setup of NEMO would produce better results. For example, assimilation of sea level data might be of substantial benefit and our current setup does not have it. Therefore, we must certainly clarify that we do not claim that HIDRA is better than NEMO in general - but rather that HIDRA does compare favorably with the only specific operational setup of NEMO at our disposal.

To make the manuscript more consistent with the referee's arguments, we have amended the manuscript to be very specific that we are referring to the specific operational setup of NEMO at Slovenian Environment Agency. We however fear that a stand-alone NEMO setup sensitivity study would diverge too far from the scope of our paper, which is to present a numerically cheap machine learning architecture which can compete with (and most often outperform) a much more complex and numerically demanding general circulation model.

**Minor comments:**

**Comment 7: How to deal with the land points in this study is missed.**

**Response:** HIDRA does not distinguish between wet and dry points - it focuses on the synoptic pattern of surface meteorological fields. It does however receive spatial encoding of atmospheric fields (x and y input channels). Again, as in topography, ECMWF land/sea mask is a constant field and as such cannot profoundly impact gradient descent during the learning process. To make this clearer, the following sentence was added to the manuscript:

*"Atmospheric fields over land and sea are treated in the same manner, i.e. while HIDRA does receive an explicit spatial encoding of atmospheric fields (Section 3.1.1), it does not employ a land/sea mask."*

**Comment 8: Why did the authors select the 29x37 for the atmospheric tensor?**

**Response:** The spatial size is specific to the input ECMWF ensemble grid. The ECMWF ensemble forecast over the domain of the study contains 73×57 grid points. As described in Section 2.3, we further downscale the data by a factor of two and end up with a grid of dimensions 37×29. Following the standard convention, spatial maps are represented as height-first tensors, thus 29×37. To clarify the origin of the atmospheric tensor dimensions we changed a line in Section 2.2 to

*"… In this study, the following forecast fields were subset to the Adriatic basin, represented by a 73×57 spatial grid (see Figure 2) …"*

and a line in Section 2.3 to

*"The data is standardised and global average pooling is used to reduce the dimensionality of the atmospheric data -- spatial dimension of the data in samples is reduced in half, from 73 x 57 points to 37 x 29 points, and the temporal dimension is reduced by a factor of 4."*

**Comment 9: It's better to give a table for the HIDRA and NEMO configuration**

**Response:** As pointed out in our response to Reviewer's Comment 5, we now include the NEMO configuration namelist in the supplementary material. Apart from the architectural design of HIDRA specified in Figure 3, the only free parameters are the learning rates and batch sizes summarized in the last paragraph of Section 3.1.4.

---

## Author Response (AR2)

Dear Topical Editor,

Thank you for your comments. We have further clarified our referral to our particular setup of the NEMO ocean model and made sure the submission adheres to the GMD publication and code standard.

With regard to the referral to the particular setup of the NEMO, we updated the Introduction and Conclusion to reflect that the drawn conclusions on HIDRA/NEMO comparioson, refer to a particular operational setup of NEMO.

The following sentences have been added to the Introduction:

*"HIDRA is benchmarked against the operational setup of NEMO v3.6 general circulation model engine, which is run daily as part of the National Hydrological Forecasting Service at the Slovenian Environment Agency (ARSO). For brevity, we refer to this particular setup (see the Code and data availability Section for a detailed configuration namelist) as the NEMO model."*

The following discussion has been added to the Conclusion:

*"HIDRA compares favorably to the current operational NEMO setup of the National Hydrological Forecasting Service. While further tuning of the operational NEMO setup at the Agency is also under way (with the aim of improving its forecasting skill), results presented in this study nevertheless indicate that HIDRA is an appropriate candidate for Slovenian Environment Agency's operational pipeline."*

To adhere to the GMD guidelines regarding the code submission, we have updated the provided source code, to include a detailed description of the training data structure and we provided a training script. We believe this should increase the reproducibility of our work and allow straight-forward training and application on third-party data. The manuscript has been updated to reference the new version of the source code.

On behalf of all the authors, best regards,
Lojze Žust

---

## Author Response (AR3)

**Authors' response to the Topical Editor**

**Comment 1: The authors' response to anonymous referee#1, Comment 2:" The authors declare "Extending the historical horizon beyond 24 hours did not significantly affect the prediction accuracy". Please give a concise description of how to find the trade-off between the model forecast accuracy and the computing resource." Could you be more specific on how to find the trade-off between the model forecast accuracy and the computing resource? This is relevant for the reader to understand your paper.**

**Response:** Thank you for the comment. Increasing the horizon increases the computational complexity, but also potentially the prediction accuracy. We selected the horizon value $T_{min}$ as the minimal value beyond which the accuracy increase was negligible. We have modified Section 4.1.1 to reflect this. We paste the changed paragraph here for convenience:

*"Increasing the length of the historic horizon defined by the parameter $T_{min}$ (see Section 3.1) increases the HIDRA execution time due to a substantially increased number of parameters in the input layer. We therefore analyzed the influence of the HIDRA historic horizon length, to find the best trade-off between the model forecast accuracy and the execution time. Table 1 summarizes the performance of HIDRA variants with $T_{min} \in \{12,24,36,48\}$, which translates to historic horizons of 12, 24, 36 and 48 hours prior to the beginning of the forecast. Prediction accuracy substantially increased by increasing the historic horizon from 12 to 24 hours (9% reduction in RMSE)but saturated for larger values. Since we have not observed measurable benefits for horizons beyond 24h, we have selected $T_{min} = 24$ as the optimal horizon value. In the remaining analysis, we denote this version as $HIDRA_0$."*

**Comment 2: The authors decided to set Tmin = 23, however, the value of Tmin in the new version5 manuscript is still 24.**

**Response:** The length of the historic horizon (24h) did not change during the review process. Already in the initial submission $T_{min} = 23$ denoted the historic horizon of 24h which turned out to be somewhat misleading. Therefore we changed this definition in the response to Reviewer #2 so that  the horizon value of 24h corresponds to the parameter value $T_{min} = 24$ in the interest of clarity and to prevent  further  confusion. This change was made consistent throughout the paper.

Kind regards,
Lojze Žust,
On behalf of the authors